

# Rarefied particle motions on hillslopes: 4. Philosophy

David Jon Furbish[1] and Tyler H. Doane[2, 3]

[1]Department of Earth and Environmental Sciences, Vanderbilt University, Nashville, Tennessee, USA
[2]Department of Geosciences, University of Arizona, Tucson, Arizona, USA
[3]Current: Department of Earth and Atmospheric Sciences, Indiana University, Bloomington, Indiana, USA

**Correspondence:** David Furbish (david.j.furbish@vanderbilt.edu)

**Abstract.** Theoretical and experimental work (Furbish et al., 2020a, 2020b, 2020c) indicates that the travel distances of rarefied particle motions on rough hillslope surfaces are described by a generalized Pareto distribution. The form of this distribution varies with the balance between gravitational heating due to conversion of potential to kinetic energy and frictional cooling by particle-surface collisions. The generalized Pareto distribution in this problem is a maximum entropy distribution constrained by a fixed energetic "cost" — the total cumulative energy extracted by collisional friction per unit kinetic energy available during particle motions. The analyses leading to these results provide an ideal case study for highlighting three key elements of a statistical mechanics framework for describing sediment particle motions and transport: the merits of probabilistic versus deterministic descriptions of sediment motions; the implications of rarefied versus continuum transport conditions; and the consequences of increasing uncertainty in descriptions of sediment motions and transport that accompany increasing length and time scales. We use the analyses of particle energy extraction, the spatial evolution of particle energy states, and the maximum entropy method applied to the generalized Pareto distribution as examples to illustrate the mechanistic yet probabilistic nature of the approach. These examples highlight the idea that the endeavor is not simply about adopting theory or methods of statistical mechanics "off the shelf," but rather involves appealing to the *style of thinking* of statistical mechanics while tailoring the analysis to the process and scale of interest. Under rarefied conditions, descriptions of the particle flux and its divergence pertain to ensemble conditions involving a distribution of possible outcomes, each realization being compatible with the controlling factors. When these factors change over time, individual outcomes reflect a legacy of earlier conditions that depends on the rate of change in the controlling factors relative to the intermittency of particle motions. The implication is that landform configurations and associated particle fluxes reflect an inherent variability ("weather") that is just as important as the expected ("climate") conditions in characterizing system behavior.

## 1 Introduction

In three companion papers (Furbish et al., 2020a, 2010b, 2020c) we examine a theoretical formulation of the probabilistic physics of rarefied particle motions and deposition on rough hillslope surfaces. As noted by Furbish et al. (2020a), such motions include the ravel of particles following disturbances (Roering and Gerber, 2005; Doane, 2018; Doane et al., 2019; Roth et al., 2020) or release from obstacles (e.g., vegetation) following failure of the obstacles (Lamb et al., 2011, 2013; DiBiase and Lamb, 2013; DiBiase et al., 2017; Doane et al., 2018, 2019), and the motions of rock fall material over the surfaces of talus



and scree slopes (Gerber and Scheidegger, 1974; Kirkby and Statham, 1975; Statham, 1976). The formulation is based on a description of the kinetic energy balance of a cohort of particles treated as a rarefied granular gas, and a description of particle deposition that depends on the energy state of the particles. The formulation predicts a generalized Pareto distribution of particle travel distances whose form varies with the balance between gravitational heating due to conversion of potential

to kinetic energy and frictional cooling by particle-surface collisions. Specifically, the generalized Pareto distribution varies from a bounded form associated with thermal collapse and rapid deposition to an exponential form representing isothermal conditions to a heavy-tailed form associated with net heating of particles and decreased deposition. As described in Furbish et al. (2020b), these varying forms of the generalized Pareto distribution are consistent with laboratory measurements of particle travel distances reported by Gabet and Mendoza (2012) and Furbish et al. (2020b), and with field-based measurements of

travel distances reported by DiBiase et al. (2017) and Roth et al. (2020). Moreover, as described in Furbish et al. (2020c), the generalized Pareto distribution in this problem is a maximum entropy distribution (Jaynes, 1957a, 1957b) constrained by a fixed energetic "cost" — the total cumulative energy extracted by collisional friction per unit kinetic energy available during particle motions. That is, among all possible accessible microstates — the many different ways to arrange a great number of particles into distance states where each arrangement satisfies the same fixed total energetic cost — the generalized Pareto

distribution represents the most probable arrangement.

The analyses of rarefied particle motions in these companion papers collectively provide an ideal case study for highlighting key elements of a statistical mechanics framework for describing sediment particle motions and transport. Indeed, as noted in the second companion paper (Furbish et al., 2020b):

> "We enjoy eating our favorite tortilla chips, and mostly we enjoy them with a well prepared dip, for example,
> 20   spicy guacamole. But... [t]he experience then is no longer about the chips, it's about the dip. The chips are just the guacamole delivery system... Similarly, these companion papers nominally concern particle motions on inclined rough surfaces. But these particles are just the delivery system. The dip consists of *the coherent statistical mechanics framework for describing the particle motions, and a demonstration that such a framework... is possible*. This represents a solid basis for... fresh ideas concerning particle motions more generally."

To wit, the purpose of this fourth companion paper is to elaborate the italicized part of the paragraph above. Specifically, we consider three framework elements: 1) the purpose and merits of probabilistic versus deterministic descriptions of particle motions; 2) the implications of rarefied versus continuum transport conditions in defining the particle flux and its divergence; and 3) the consequences of increasing uncertainty in descriptions of sediment transport that accompany increasing length and time scales.

We suggest that the timing is ideal for offering perspective on these elements of a statistical mechanics framework. Amidst echoes from the pioneering work of Einstein (1937) on bed load particle motions and that of Culling (1963) on hillslope soil creep, there is a reemerging interest in probabilistic descriptions of sediment motions and transport. For example, recent efforts involve descriptions of: 1) bed load particle motions and transport; 2) bed load tracer particle motions, including effects of particle-bed exchanges; 3) nonlocal sediment transport on hillslopes; 4) particle motions in soils, including tracer particles;





and 5) rain splash transport (Appendix A). However, this effort is a patchwork of approaches and methods, and to date it mostly has involved kinematic descriptions of motions and transport with limited elucidation of the associated mechanics. We believe that it is important for the philosophical underpinning of this growing effort to be part of the conversation, adding to recent perspectives on bed load transport offered by Ancey (2020a, 2020b). This includes attention to commonalities in the formalism used to describe transport in different settings, for example, in relation to transport on hillslopes and within rivers. The conversation also must include an honest assessment of the expectations and limitations of probabilistic descriptions of transport.

In Section 2 we summarize key material from the three companion papers for reference in later sections. In Section 3 we step back and consider in general terms the philosophical basis of a statistical mechanics framework for describing sediment motions and transport. In Section 4 we return specifically to the problem of rarefied particle motions on hillslopes and use the analysis to illustrate elements of the framework. In Section 5 we consider implications of the statistical mechanics description of rarefied particle motions on hillslopes. In several sections we provide historical background on the technical material covered.

## 2 Background

With reference to material in Furbish et al. (2020a, 2020b, 2020c), the problem of describing rarefied particle motions on hillslopes is motivated by the entrainment forms of the flux and the Exner equation. Namely, let $f_r(r; x, t)$ denote the probability density function of the travel distances $r$ of particles whose motions start at $x$, and let $R_r(r; x, t)$ denote the associated exceedance probability function. Assuming motions are only in the positive $x$ direction and noting that $x' = x - r$, the particle volumetric flux $q(x, t)$ may be written as (Furbish and Roering, 2013; Furbish et al., 2017)

$$q(x,t) = \int\limits_{-\infty}^{x} E_s(x',t) R_r(x - x'; x', t)\, \mathrm{d}x', \qquad (1)$$

where $E_s(x, t)$ denotes the particle volumetric entrainment rate at position $x$ and time $t$. In turn, letting $\zeta(x, t)$ denote the local land-surface elevation, the entrainment form of the Exner equation is (Tsujimoto, 1978; Nakagawa and Tsujimoto, 1980)

$$c_b \dot{\zeta}(x,t) = c_b \frac{\partial \zeta(x,t)}{\partial t} =$$

$$-E_s(x,t) + \int\limits_{-\infty}^{x} E_s(x',t) f_r(x - x'; x', t)\, \mathrm{d}x', \qquad (2)$$

where $c_b = 1 - \phi_s$ is the particle volumetric concentration of the surface with porosity $\phi_s$. The central elements of Eq. (1) and Eq. (2) are the probability density function $f_r(r; x, t)$ and the associated exceedance probability function $R_r(r; x, t)$. These are related to the disentrainment rate function defined by

$$P_r(r; x, t) = \frac{f_r(r; x, t)}{R_r(r; x, t)}, \qquad (3)$$





which, when multiplied by $dr$, is interpreted as the probability that a particle will become disentrained within the small interval $r$ to $r+dr$, given that it "survived" travel to the distance $r$. The disentrainment rate, Eq. (3), connects the descriptions of the flux and the rate of change in the land-surface elevation, Eq. (1) and Eq. (2), to the physics of particle motions and disentrainment. We also note that Eq. (1) and Eq. (2) are nonlocal in form and scale independent.

5     The appearance of time $t$ in Eq. (1) and Eq. (2) is examined further below. Suffice it to say here that this is in reference to time variations associated with ensemble expected values of the flux $q(x,t)$ and the rate $\dot{\zeta}(x,t)$, or to variations in appropriately defined time averages of these quantities (Furbish and Haff, 2010). In addition we are neglecting particle travel times throughout. For reference below the left side of Eq. (2) also may be written in terms of the divergence form of the Exner equation, namely, $c_b\dot{\zeta}(x,t) = c_b\partial\zeta(x,t)/\partial t = -\partial q(x,t)/\partial x$ when the flux $q(x,t)$ is specified by Eq. (1) (Appendix B).

10     The analysis presented in Furbish et al. (2020a) describes the mechanical basis of the disentrainment rate $P_r(r;x,t)$ and the associated probability distribution $f_r(r;x,t)$. This involves a consideration of the kinetic energy balance of rarefied particle motions and how this balance determines the deposition of particles in relation to their energy state. In particular the analysis leads to the result that for a given particle size and shape the disentrainment rate on an inclined surface with uniform slope and roughness is

$$15 \quad P_r(r;x,t) = \frac{1}{Ar+B}, \tag{4}$$

which in turn leads to the generalized Pareto distribution,

$$f_r(r;x,t) = \frac{B^{1/A}}{(Ar+B)^{1+1/A}}, \tag{5}$$

where $A \in \Re$ is a shape parameter and $B > 0$ is a scale parameter (Pickands, 1975; Hosking and Wallis, 1987) (Figure 1). The

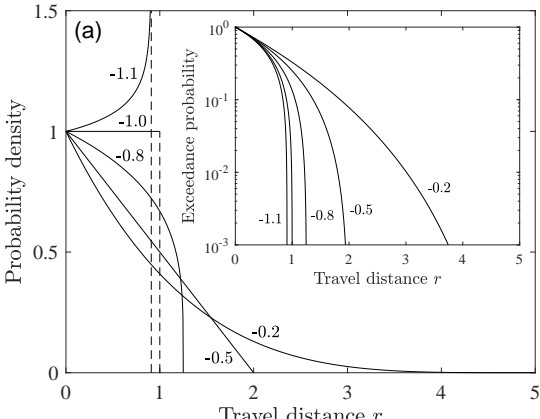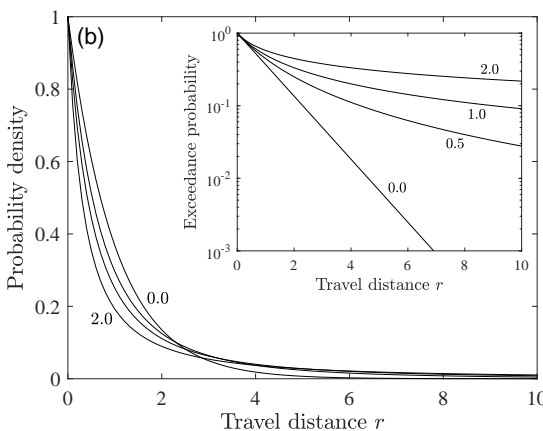

**Figure 1.** Plot of probability density $f_r(r;x)$ versus travel distance $r$ for scale parameter $B = 1$ and different values of the shape parameter $A$ for (**a**) $A < 0$ and (**b**) $A \geq 0$ with associated exceedance probability plots (insets). Figure reproduced from companion paper (Furbish et al., 2020a). Compare with Figure 1 in Hosking and Wallis (1987).





cumulative distribution is

$$
F_r(r;x,t) = \begin{cases} 1 - \dfrac{B^{1/A}}{(Ar+B)^{1/A}} & A \neq 0 \\ 1 - e^{-r/B} & A = 0, \end{cases}
\tag{6}
$$

and the exceedance probability is

$$
R_r(r;x,t) = \begin{cases} \dfrac{B^{1/A}}{(Ar+B)^{1/A}} & A \neq 0 \\ e^{-r/B} & A = 0. \end{cases}
\tag{7}
$$

For $A < 1$ the mean is

$$
\mu_r = \frac{B}{1-A},
\tag{8}
$$

and for $A < 1/2$ the variance is

$$
\sigma_r^2 = \frac{B^2}{(1-A)^2(1-2A)}.
\tag{9}
$$

The mean is undefined for $A \geq 1$ and the variance is undefined for $A \geq 1/2$. Note that because the density $f_r(r;x,t)$ may vary (slowly) with time $t$, the parameters $A$ and $B$ also may vary with time.

In mechanical terms the shape and scale parameters $A$ and $B$ are

$$
A = \frac{\alpha}{\gamma}\left[\frac{S}{\mu} - 1 + \frac{1}{\alpha}(\gamma - 1)\right] \qquad \text{and}
\tag{10}
$$

$$
B = \frac{\alpha}{\gamma}\frac{E_{a0}}{mg\mu\cos\theta}.
\tag{11}
$$

Here, $S$ is the magnitude of the slope inclined at an angle $\theta$, $m$ is particle mass, $g$ is acceleration due to gravity, $\mu$ is a friction factor due to extraction of particle kinetic energy $E_p = (m/2)u^2$ where $u$ is the surface-parallel particle velocity, $E_a = \langle E_p \rangle$ is the arithmetic average particle energy so that $E_{a0}$ is the initial average energy at $r = 0$, $\gamma = E_a/E_h$ where $E_h$ is the harmonic average particle energy, and $\alpha = \alpha_0/(1 - \mu_1 Ki)$ where $\alpha_0$ and $\mu_1$ are factors of order unity and $Ki$ is the Kirkby number defined by

$$
Ki = \frac{S}{\mu},
\tag{12}
$$

which represents the ratio of gravitational heating to frictional cooling. Here we emphasize that $mg\cos\theta$ in Eq. (11) is not to be interpreted as the static normal weight of the particle, and $\mu$ is not interpreted as a Coulomb-like friction coefficient. Rather, $\mu \sim \langle \beta_x \rangle$, where $\langle \beta_x \rangle$ denotes the expected proportion of particle kinetic energy extracted per particle-surface collision during downslope motion. Details are provided in Furbish et al. (2020a, 2020b).





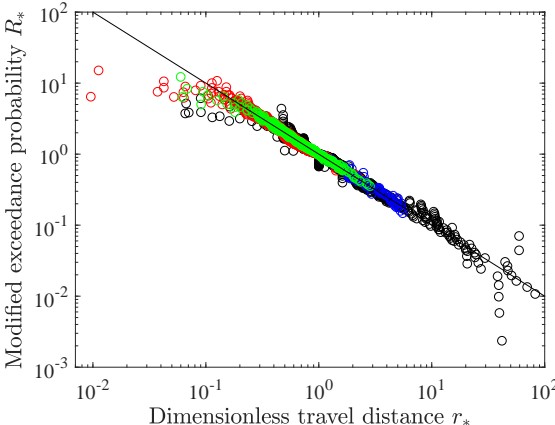

**Figure 2.** Plot of modified exceedance probability $R_*$ versus dimensionless travel distance $r_*$ and line with log-log slope of -1 for laboratory experiments described by Gabet and Mendoza (2012) (green) and Furbish et al. (2020b) (red) and field-based experiments described by DiBiase et al. (2017) (blue) and Roth et al. (2020) (black). Data for $A < 0$ fall to left of $r_* = 10^0 = 1$ with values in the tails represented by smaller values of $r_*$. Data for $A > 0$ fall to the right of $r_* = 10^0 = 1$ with values in the tails represented by larger values of $r_*$. Total data number is $N = 5671$. Figure reproduced from Furbish et al. (2020c).

Following Furbish et al. (2020b) we calculate the quantities

$$R_* = R_r^A \quad \text{and} \quad r_* = \frac{A}{B}r + 1. \tag{13}$$

Based on Eq. (7), values of the modified exceedance probability $R_*$ and the dimensionless travel distance $r_*$ should collapse to a straight line in a log-log plot with slope of -1 (Figure 2). The data in this figure, spanning more than three orders of magnitude

of the dimensionless travel distance $r_*$, are compiled from Furbish et al. (2020b; Figure 16 therein). Values of $A$ and $B$ are estimated from laboratory measurements of particle travel distances reported by Gabet and Mendoza (2012) and Furbish et al. (2020b), and from field-based measurements of travel distances reported by DiBiase et al. (2017) and Roth et al. (2020). This plot does not prove, but nonetheless supports, the idea that the generalized Pareto distribution correctly describes the energetics of the behavior of rarefied particle motions for a variety of slope and surface roughness conditions. The data fits for individual

experiments with detailed explanation are presented in Furbish et al. (2020b).

We refer to elements of the analysis summarized here in several sections below. Meanwhile we turn to the philosophy of the statistical mechanics framework.

## 3 Philosophy of the statistical mechanics framework

Although our companion papers are focused on rarefied particle motions on hillslopes, here we purposefully step back and ini-

15 tially consider the broader topic of probabilistic descriptions of sediment motions and transport. Borrowing ideas and wording from a book in preparation, we briefly consider elements of three foundational concepts in the natural sciences that repeatedly





appear in the study of complex systems, focusing here on sediment systems: 1) the relation between mechanistic descriptions of sediment behavior and probabilistic versus deterministic formulations of this behavior; 2) differences in rarefied versus continuum conditions; and 3) the treatment of uncertainties in descriptions of system behavior that grow with increasing length and time scales considered. The point of this brief overview is to ask ourselves, at least momentarily, to step out of our comfort

zones as informed and conditioned by our different backgrounds in, say, particle mechanics, continuum mechanics, probability and statistics, and, by the length and time scales with which we are most familiar. Following this overview, we return to the problem of rarefied particle motions and use it to illustrate the philosophical points involved.

## 3.1 Probabilistic versus deterministic descriptions

In learning how to describe the behavior of mechanical systems, mostly we are initially exposed to deterministic examples. We

study Newton's laws as these pertain to simple particle systems, and then move on to the behavior of solids and fluids treated as continuous materials, wrapping our heads around Lagrangian versus Eulerian perspectives. The formalism is unambiguous, and describing the behavior of a well constrained system is in principle straightforward. Indeed, much of the legacy of geophysics resides in the determinism of continuum mechanics. Perhaps it is therefore natural that we might envision that a mechanistic description of the behavior of a system implies that such a description ought to be, or perhaps only can be, a deterministic

one. Such a perception represents a lost opportunity. The most elegant counterpoint example is the field of classical statistical mechanics — devoted specifically to the probabilistic (i.e., non-deterministic) treatment of the behavior of gas particle systems in order to justify the principles of thermodynamics — yet which is no less mechanical in its conceptualization of this behavior than, say, the application of Newton's laws to the behavior of a deterministic system consisting of the interactions of a few billiard balls, or involving the motion of a Newtonian fluid subject to specific initial and boundary conditions.

Once steeped in the language of mechanics, we understandably take comfort in mechanistic descriptions of system behavior. Specifically, we invest trust in the underlying foundation, and implied rigor, provided by classical mechanics. This is a good thing. But given the complexity and the uncertainty in describing the behavior of sediment systems, here it is essential to consider the idea that the concepts and language of probability are well suited to the problem of describing this behavior — precisely because of the complexity and uncertainty involved. This involves relaxing our expectations, for example, that a

deterministic-like relationship exists between the flux of bed load sediment and the fluid stress imposed on the streambed, or between the flux of sediment on a hillslope and the local land-surface slope — particularly when these involve noise driven processes, as described below. This idea of leaning on probability to describe the behavior of sediment systems is not as straightforward as describing the behavior of idealized gas particle systems. Nonetheless, the objective is the same: to be mechanistic, yet probabilistic. These worldviews are entirely compatible.

To be sure, the extent to which the tools of probability can be fruitfully brought to bear to characterize particle motions and transport vary with the specific process considered, and the information we have available to constrain any particular probabilistic description of motions. For example, we know far more about the probabilistic qualities of bed load sediment transport in shear flows based on flume experiments than, say, soil particle transport and mixing associated with bioturbation and granular creep (Appendix A). The objective therefore is to aim at probabilistic descriptions of sediment particle motions





and transport that lean on the *style of thinking* of statistical mechanics, recognizing that this endeavor is not simply about adopting established theory or methods "off the shelf." Rather, such efforts involve tailoring descriptions of transport to the process, the scales of interest and the techniques of observation and measurement used. The examples covered below illustrate these points.

## 3.2 Rarefied versus continuum conditions

The continuum hypothesis — the essential basis of continuum mechanics — stands as a triumph of the physical sciences. (Let us be clear that we are referring to the version of this hypothesis as applied to descriptions of real material systems rather than to the related mathematical idea posed by Georg Cantor, that there is no set of numbers whose size falls between the two infinities associated with the natural numbers and the real numbers.) This hypothesis allows us to envision many solid and fluid materials at our ordinary macroscopic scale of observation as being continuous things whose properties and behavior can be described using that part of the calculus given to continuously differentiable functions — even though when we focus our attention on the scale of the elements of a 'continuous' material, that is, at the particle scale, we discover that it is decidedly discontinuous. Indeed, many of the definitions of basic, familiar quantities describing the properties, rheology and motion of real materials — their intensive properties, thermodynamic state variables, rheological coefficients, discharges and fluxes, the divergence of these fluxes, etc. — at the outset assume continuous substances and continuum behavior that involve smooth changes with respect to space and time. That said, this lovely continuum siren is to be avoided as a *de facto* starting point in descriptions of sediment motions and transport. Many, if not most, sediment particle motions on Earth's surface are patchy, intermittent and demonstrably rarefied (Furbish et al., 2016b, 2018c; Ancey, 2020a, 2020b) — conditions that are at odds with continuum formulations of these motions.

For these reasons an appropriate strategy involves constructing descriptions of the collective behavior of sediment particles without assuming a continuum behavior at the outset. Indeed, precise definitions of the sediment particle flux and its divergence do not assume continuum conditions (Ancey, 2010; Furbish et al., 2012a, 2016b, 2017; Ancey and Pascal, 2020). Instead, the idea is to develop more general (probabilistic) formulations of this behavior, and then ask the question: Under what conditions does a continuum formulation of behavior make sense? As a point of reference, when continuum behavior is assumed at the outset, the Navier-Stokes momentum equation is derived from the Cauchy momentum equation. But when viewed with respect to particle (molecular) behavior, the Navier-Stokes equation is derived from the Boltzmann equation — which is decidedly probabilistic and entirely agnostic to continuum versus rarefied conditions. That is, the Boltzmann equation is equally applicable to both conditions. If the continuum hypothesis is satisfied, then it is natural to adopt the Navier-Stokes formalism. On the other hand, rarefied conditions must be treated probabilistically using methods of statistical mechanics. As described below with respect to sediment particles, this can include the use of continuum-like equations — noting that "continuum-like" means continuously differentiable, not that the particles behave as a continuum, and also noting that such equations apply to ensemble expected conditions, not to individual realizations. This means that we must be careful in interpreting the use of continuous probability distributions and related functions to describe attributes of particle motions (e.g., entrainment rates, travel distances, etc.) as in Eq. (1) and Eq. (2).



One of the most important consequences of rarefied transport conditions is this: One cannot expect to predict a well-defined single value of the particle flux from specified, fixed controlling factors. Even under the ideal circumstances of a "perfect" model of the particle flux, such a prediction must be probabilistic. That is, a given set of controlling factors yields a probability distribution of fluxes rather than a single value. Any individual realization therefore can involve a value that may or may not

coincide with a statistically expected value, whether this expected value is an empirical outcome or is predicted by a mechanical argument.

### 3.3   Uncertainty with growing scales

Our interest in sediment particle systems spans timescales of less than milliseconds to hundreds of thousands of years. The shortest timescales are represented by, say, observations of the details of particle motions in controlled experiments measured

by high-speed imaging. Intermediate timescales are represented by, say, measurements of transport on hillslopes and in rivers on human timescales pertaining to the erosion and deposition of sediment in relation to land-use and river management. Long timescales are represented by our interest in understanding the evolution of hillslope and river systems at geomorphic timescales. Similarly, our interest in sediment systems spans length scales of less than a millimeter to at least tens or hundreds of kilometers. The smallest length scales are represented by differential particle motions during granular creep that are a fraction

of a particle diameter, or in relation to the initial jiggling of bed load particles prior to entrainment from their microtopographic 'pockets.' Intermediate length scales are represented by particle motions involved in the dynamics of river and aeolian bedforms — ripples to dunes to megadunes — thence to scales involving, say, intermittent sediment motions from the crest of a hillslope to its base or within the extent of one or two river bends. The largest length scales are represented by the erosion and deposition of sediment over the scale of a hillslope-channel network or a depositional basin.

With increasing scale (length and time) goes increasing uncertainty in our descriptions of sediment motions and the behavior of sediment systems. The essential reasons for this increasing uncertainty reside in the increasing complexity, including heterogeneity, of sediment systems as their size increases, and in the increasing stochasticity, including the patchiness and intermittency, of factors that influence sediment motions and transport as both the system size and the time scale of interest increase. Equally important, with increasing scale our uncertainty grows in relation to the increasing difficulty, and the loss

of resolution, associated with observing and measuring quantities that enter into our descriptions of sediment motions and transport — whether these quantities involve features of the sediment itself (e.g., particle sizes and arrangements, details of particle motions), or attributes of the factors influencing sediment transport (e.g., changing fluid motions, surface roughness). Moreover, in approaching climate-change time scales and longer, we can only imagine in probabilistic terms how many of the ingredients of sediment transport might vary (Benda and Dunne, 1997).

In relation to the uncertainty that grows with scale, we also must consider the consequences of differences in our ability to observe and measure quantities representing the dynamics of "fast" versus "slow" systems as viewed relative to the human experience. Focusing specifically on the configuration of a sediment system — a bedform, a river reach, a soil mantled hillslope — a fast system is one for which we can observe and measure attributes of the particle fluxes and associated changes in the system configuration over human time scales. A slow system is one for which the fluxes and changes in configuration are largely



imperceptible over these time scales. In simple terms, for a system consisting of $N$ particles whose configuration changes due to a characteristic particle flux $q$ [T$^{-1}$], the ratio $T_r \sim N/q$ is akin to a particle residence time. In turn, let $T_e$ denote a characteristic observation time — an experimental run time, the duration of a research project, the time record of satellite images, a researcher's career. Then the ratio $T_e/T_r$ is a rough measure of our capacity to observe, although not necessarily

measure, the "completeness" of the dynamics of the system, that is, its full dynamical behavior. Note that individual particle motions might be fast, but their contribution to changes in system configuration may be slow, as in the case of rarefied particle transport on hillslopes. From this perspective the stark contrasts between the capabilities and strategies of studies of transport in hillslope and river systems and their evolution become clear. For example, the ability to create a small version of a river reach in a laboratory, then measure long time series of bed load flux in order to fully characterize the fluctuations and ensemble

averaged behavior of such series in relation to bedform dynamics (Dhont and Ancey, 2018; Ancey and Pascal, 2020) is simply not a possibility in studies of natural soil creep and its long term consequences. Indeed, we have only recently achieved the ability to measure small particle motions involved in soil creep (Deshpande et al., 2020), yet the particle residence times of soil mantled hillslopes may be 10,000 years or more, thus requiring indirect measures of particle behavior such as tracer particle mixing (Furbish et al., 2018c; Gray et al., 2020).

15                                                         **********

These ideas support a strong case for incorporating concepts and methods of probability — the natural language of un-certainty — into our descriptions of sediment particle motions and transport, tuning the specifics to the demands of different scales. This is as much a philosophical choice as a technical one; it is a choice to make the treatment of uncertainty a key part of the problem at the outset (Ancey and Pascal, 2020; Korup, 2020). Of course the strategies and methods vary with scale, as do

the sources of uncertainty, in relation to the transport processes involved and the techniques of observation and measurement used. The purpose of explicitly incorporating probabilistic concepts in describing transport is to use this as a framework to explore, for example, the consequences of patchiness and intermitency of sediment motions in formulations of transport rates; or how a predicted transport rate at a specified position and time within a real system actually represents a statistically expected behavior associated with a distribution of possible transport rates. The objective is to illustrate that this approach to uncertainty,

combined with aiming at mechanistic, albeit probabilistic, descriptions of sediment particle behavior — avoiding a continuum description at the outset — will move us toward a deeper understanding of sediment particle motions and transport in both experimental and natural systems. We now step through the elements of the probabilistic framework outlined in the preceding sections with specific reference to the problem of rarefied particle motions on hillslopes.

## 4   Elements of the framework

### 4.1   Probabilistic versus deterministic descriptions

Here we consider three elements of the formulation presented in Furbish et al. (2020a, 2020b, 2020c) to highlight the purpose and merits of a probabilistic description of particle motions and disentrainment. The first concerns our treatment of the extrac-





tion of kinetic energy of a particle during particle-surface collisions as a random variable versus appealing to the deterministic idea of fixed coefficients of restitution. The second concerns our use of the Fokker-Planck equation to describe the changing energy states of the particles during their downslope motions, leading to deposition, versus considering only average particle energy conditions. The third concerns our efforts to demonstrate that the generalized Pareto distribution in this problem is a

maximum entropy distribution. The objective is to highlight the mechanistic yet probabilistic nature of the analyses.

### 4.1.1 Particle energy extraction

We start with some background. In classical statistical mechanics the starting point for describing the motions and collective behavior of particles undergoing conservative collisions is the Boltzmann equation. This equation describes the evolution of the joint probability density function of particle positions and velocities in relation to the forces acting on the particles.

Depending on the formulation of particle collisions (e.g., Chapman–Enskog theory), the Boltzmann equation leads to the Navier-Stokes equation in the continuum limit of vanishing Knudsen number. For dissipative collisions in granular materials, however, one must incorporate effects of energy losses during particle collisions. In pioneering work, Haff (1983) formulated the hydrodynamic analogue of the Navier-Stokes equations for granular flows. In this formulation he envisioned simple inelastic collisions and appealed to the normal coefficient of restitution to characterize energy losses during collisions, neglecting the

details of particle collisions in the scalar treatment. The thermodynamic temperature is replaced with the granular temperature, and the hydrodynamic equations are supplemented with the mechanical energy equation in order to characterize granular flow behavior. One of the key outcomes of this work is Haff's cooling law (Brito and Ernst, 1998; Nie et al., 2002; Brilliantov and Pöschel, 2004; Dominguez and Zenit, 2007; Brilliantov et al., 2018; Yu et al., 2020), which predicts that when the external source of energy is removed the granular temperature decays with time as $\sim t^{-2}$ in the homogeneous cooling state for a velocity

independent coefficient of restitution (Brilliantov and Pöschel, 2005; Yu et al., 2020).

Now consider particle-surface collisions in the rarefied particle motion problem. Of interest are downslope motions and travel distances. The energy balance described in Furbish et al. (2020a) thus focuses on the particle kinetic energy state $E_p = (m/2)u^2$ involving the surface-parallel velocity $u$. Energy is extracted during particle-surface collisions, and analogous to collisions in the granular gas problem we define the proportion of energy extracted during a collision as

$$\beta_x = -\frac{\Delta E_p}{E_p} \, . \tag{14}$$

As noted by Williams and Furbish (2020):

> "The quantity $\beta_x$ is nominally related to a coefficient of restitution $\epsilon_x$ as $\beta_x = 1 - \epsilon_x^2$. However, the change in translational energy $\Delta E_p$ is partitioned between deformational friction, rotational energy and transverse motion, so the coefficient $\epsilon_x$ (and therefore the factor $\beta_x$) cannot simply represent a coefficient of restitution — although
>
> 30     particle collision theory suggests that this coefficient includes effects of normal and tangential coefficients of restitution as normally defined (Brach, 1991; Stronge, 2000). This means that $\beta_x$ must be treated formally as a random variable rather than a fixed deterministic quantity as in granular gas theory."





Indeed, unlike idealized conditions often envisioned in collision theory, moving particles "see" a rough irregular surface rather than a smooth planar surface such that, for any incident trajectory angle measured relative to the $x$ axis, the actual incident collision angle may vary significantly; this includes angular incidence measured in the transverse direction depending on the local configuration of the surface. The geometrical irregularity of natural particles further increases the geometrical complexity

of possible collisions, and collision histories during downslope motion are unique and highly variable. Rather than attempting to consider the mechanical details of these motions and collisions in a deterministic manner (see Appendix E in Furbish et al. 2020a), it instead becomes defensible to more simply pool the partitioning of energy into different forms and treat this energy extraction as a random variable, as in Eq. (14). In effect we are asking ourselves to blur our eyes to the myriad details of surface roughness texture, particle shape, particle trajectories and collision mechanics, and instead aim at a granularity suited to the

task. This is not dissimilar to granular gas theory in which details of collisions are neglected, and energy dissipation is assumed to be adequately characterized by a single coefficient of restitution, either fixed or velocity dependent.

With this description of particle energy extraction in place, it then becomes straightforward to characterize the number of particle-surface collisions per unit distance and in turn the collisional energy loss per unit distance in relation to the surface-parallel velocity $u$ (Furbish et al. 2020a). These descriptions are entirely analogous to the particle collision frequency and the

rate of energy loss due to collisions in granular flows, as described by Haff (1983). This provides the basis for defining the characteristic deposition length as described in the next section.

What might an alternative approach look like? One possibility involves using discrete element methods (DEMs) to directly mimic particle motions on rough surfaces, extracting information from the simulations to characterize particle-surface collisions and energy extraction. Such simulations essentially represent numerical analogues of physical experiments. An obvious

advantage is speed in examining different conditions, for example, surface slopes, roughness configurations and particle sizes, using the motions of a great number of particles versus the relatively small numbers of particles used in experiments. Add to this the capability to readily extract information on details of motions and collisions that are not accessible in physical experiments, except possibly via high-speed imaging. Disadvantages include the difficulty of mimicking irregular particle shapes and creating realistic surface-roughness textures. (And let us note that informed use of DEMs, for example LAMMPS

and LIGGGHTS, is not a plug-and-play endeavor.) Nonetheless, one could potentially learn much in a generic sense about particle-surface interactions from DEMs, particularly if conducted in concert with carefully designed physical experiments. This includes assessing how sensitive macroscopic measures of particle motions (e.g., travel distances) are to variations in controlling factors — for example, particle shape and roughness texture — as these factors are varied. But rather than imagining the need to mimic all details of realistic conditions associated with a hillslope surface prototype, simulations should be

designed to examine particle-surface interactions in a manner that allows for generalization of elements of collisional friction.

Herein arises a need for pause in using DEMs. Namely, these methods can quickly generate enormous amounts of numerical information on the details of particle motions and collisions. At risk is using a big numerical hammer to pound on the problem, mimicking particle motions without regard to elucidating general underlying principles of particle behavior, defaulting to descriptions of outcomes without gaining a deeper understanding of the systematics producing them — for example, without

learning the mechanical basis of the deposition length scale that sets the pattern of deposition (Section 4.1.2); or without





learning the form of the distribution of travel distances, the mechanical basis of its parametric values, or its maximum entropy properties (Section 4.1.3). This speaks to the relevance of the cliché that much of the power of numerical simulations resides in their use in concert with theory and experiments (Emanuel, 2020). Indeed, the success of numerical simulations that have revolutionized the study of granular gas dynamics grows directly from the grounding of this topic in kinetic and statistical

mechanics theory (e.g., Brilliantov and Pöschel, 2004), which both motivates and guides the design of numerical treatments of granular gases.

### 4.1.2    Energy states and the Fokker-Planck equation

Consider a second element of the formulation, which concerns our use of the Fokker-Planck equation to describe the changing energy states of the particles during their downslope motions, leading to deposition, versus considering only average particle

energy conditions. We start with some background.

The Fokker-Planck equation represents a triumph of classical statistical mechanics. Although originally developed to describe the time evolution of the distribution of velocities of particles subjected to viscous forces and random forces associated with collisions, the name of this equation now is more generally associated with other observable quantities whose distributions evolve according to an equation having the same form. For example, with reference to the distribution of particle positions rather than velocities, the Fokker-Planck equation historically is referred to as the Smoluchowski equation. It also is referred to

as the Komolgorov forward equation in the context of Markov processes. Although the Fokker-Planck equation can be derived in several ways, perhaps the most general treatment starts with a Master equation, a general probabilistic expression of conservation of probability associated with observable states. Like the Fokker-Planck equation, there are several versions of the Master equation, sometimes referred to as the Chapman-Komolgorov equation, depending on the field and application. Here

we focus on a continuous form of the Master equation as described by Chandrasekhar (1943) and Risken (1984). We start with two familiar examples to develop the essential concepts before turning to the unfamiliar problem of rarefied particle motions addressed in Furbish et al. (2020a). Our objective is to illustrate the statistical mechanics framework of the analysis.

Let $f_x(x,t)$ denote the probability density function of particle positions $x$ at time $t$. In turn let $r = x(t + \mathrm{d}t) - x(t)$ denote a small particle displacement during the interval $\mathrm{d}t$, and let $f_r(r;x,t)$ denote the probability density function of displacements

$r$ occurring during this interval. At time $t + \mathrm{d}t$ the density of particle positions $x$ is then given by

$$f_x(x,t+\mathrm{d}t) = \int\limits_{-\infty}^{\infty} f_r(r;x-r,t)f_x(x-r,t)\,\mathrm{d}r. \tag{15}$$

This is one form of the Master equation. It says that the probability density of particles at position $x$ at time $t + \mathrm{d}t$ involves particle displacements $r$ during $\mathrm{d}t$ to this position $x$ from all possible starting locations $x - r$ as well as motions away from $x$. This expression is nonlocal and scale independent. It is examined in more detail by Furbish et al. (2012a) with respect to bed

load particle motions.

Now, assuming the density $f_r(r;x,t)$ is peaked near $r = 0$ with finite first and second moments, we may expand the integrand in Eq. (15) as a Taylor series to second order (referred to as a Kramers-Moyal expansion), subtract $f_x(x,t)$ from both sides,



then divide by $\mathrm{d}t$ and take the limit as $\mathrm{d}t \to 0$ to obtain a Fokker-Planck equation (or the Smoluchowski equation), namely,

$$\frac{\partial f_x(x,t)}{\partial t} = -\frac{\partial}{\partial x}[k_1(x,t)f_x(x,t)]$$

$$+\frac{\partial^2}{\partial x^2}[k_2(x,t)f_x(x,t)]. \tag{16}$$

In the language of statistical mechanics, $k_1(x,t) = \lim_{\mathrm{d}t \to 0}(1/\mathrm{d}t)\langle r \rangle$ [L T$^{-1}$] is a drift coefficient or drift speed, and is physi-
cally interpreted as the ensemble-averaged particle velocity. The quantity $k_2(x,t) = \lim_{\mathrm{d}t \to 0}(1/2\mathrm{d}t)\langle r^2 \rangle$ [L$^2$ T$^{-1}$] is a diffu-
sion coefficient equal to one-half the rate of change in the variance of particle positions, namely, a diffusivity. The diffusive
term in Eq. (16) is a mathematically local approximation of the nonlocal behavior embodied in the integral form of the Master
equation, Eq. (15). Note that the Fokker-Planck equation, Eq. (16), is like an ordinary advection-diffusion equation, although
it describes the time evolution of the probability distribution $f_x(x,t)$ of particle positions $x$. This form of the Fokker-Planck
equation is the basis of numerous descriptions of sediment particle transport, including tracer particles, in rivers and soils and
by rain splash (see Appendix A).

Here is a particularly important sidebar. As probabilistic expressions the Master equation and the Fokker-Planck equation
are entirely agnostic to continuum versus rarefied conditions; they are equally applicable to both. If in an individual system
(realization) the continuum hypothesis is satisfied — a condition that is independent of the probabilistic basis of the Master
equation or the Fokker–Planck equation — then the probabilistic formulation based on ensemble-expected conditions and its
continuum counterpart are essentially one and the same. If, however, the continuum hypothesis is not satisfied, then one must
appeal to a probabilistic formulation of ensemble-expected conditions (in order to justify the use of continuously differentiable
equations), with the proviso that any prediction of the behavior of an individual (rarefied) system is probabilistic in nature. In
other words, using the Fokker-Planck equation, Eq. (16), to describe the behavior of a system under rarefied conditions is in
effect the same as using a continuum-like equation, where "continuum-like" means continuously differentiable, not that the
particles behave as a continuum. Because of the significance of these points, we have reproduced in Appendix C key material
from Appendix A presented in Furbish et al. (2018c). We return to the idea of rarefied versus continuum conditions in Section
4.2.

Now let $f_u(u,t)$ denote the probability density function of particle velocities states $u$ at time $t$. In turn let $q = u(t+\mathrm{d}t) - u(t)$
denote a small change in the velocity state during the interval $\mathrm{d}t$, and let $f_q(q;u,t)$ denote the probability density function of
the changes $q$ occurring during this interval. At time $t + \mathrm{d}t$ the density of particle velocity states $u$ is then given by a Master
equation having the same form as Eq. (15). Again assuming the density $f_q(q;u,t)$ is peaked near $q = 0$ with finite first and
second moments, we follow the developments above to obtain a Fokker-Planck equation describing the time evolution of the
density $f_u(u,t)$ over the velocity domain, namely,

$$\frac{\partial f_u(u,t)}{\partial t} = -\frac{\partial}{\partial u}[k_1(u,t)f_u(u,t)]$$

$$+\frac{\partial^2}{\partial u^2}[k_2(u,t)f_u(u,t)]. \tag{17}$$



Now the drift coefficient $k_1(u,t) = \lim_{dt \to 0}(1/dt)\langle q \rangle$ [L T$^{-2}$] is interpreted as an acceleration, or the ensemble-averaged force per unit particle mass acting on particles with velocity $u$. The diffusion coefficient $k_2(x,t) = \lim_{dt \to 0}(1/2dt)\langle q^2 \rangle$ [L$^2$ T$^{-3}$] is the rate of change in the variance of velocity fluctuations about $u$, akin to a change in kinetic energy. Note that in going from a Fokker-Planck equation involving particle positions, Eq. (16), to one involving particle velocities, Eq. (17), the drift

coefficient $k_1$ transitions from a velocity to an acceleration, and the diffusion coefficient $k_2$ transitions from a rate of change in the variance of positions to a rate of change in the variance of velocities.

As a point of reference, Furbish et al. (2012b) start with the Fokker-Planck equation given by Eq. (17) to examine the statistical equilibrium distribution of the streamwise velocities of bed load particles. Wu et al. (2020) elaborate this idea by demonstrating that a large proportion of long particle hops experiencing relatively large velocities "results in a Gaussian-

like velocity distribution, while a mixture of both short and long hop distance particles leads to an exponential-like velocity distribution."

With these ideas in place, we now turn to the problem of rarefied particle motions on rough hillslopes. Here is where we highlight the idea that the endeavor is not simply about adopting theory or methods "off the shelf," but rather involves appealing to the style of thinking of statistical mechanics while tailoring the description to the process.

Because particle motions in this problem are highly rarefied and intermittent, we appeal to the idea of a cohort of particles — a Gibbs-like ensemble of systems, each containing one particle that is subjected to the same physics during downslope motion (Appendix B in Furbish et al., 2020a). Moreover, because the formulation is centered on particle travel distances and therefore on deposition over space rather than time, it aims at describing the evolution of the ensemble distribution of particle energy states with respect to downslope position $x$. Thus, let $f_{E_p}(E_p, x)$ denote the probability density function of particle energy

states $E_p$ at position $x$. In turn let $q = E_p(x + dx) - E_p(x)$ denote a small change in the energy state over the interval $dx$, and let $f_q(q; E_p, x)$ denote the probability density function of the changes $q$ occurring over this interval. At position $x + dx$ the density of particle energy states $E_p$ is then given by a Master equation having the same form as Eq. (15), where the particle position (state) $x$ is replaced with the energy state $E_p$ and time $t$ is replaced with position $x$ (Appendix C in Furbish et al., 2020a). Again assuming the density $f_q(q; E_p, x)$ is peaked near $q = 0$ with finite first and second moments, we follow the

developments above to obtain a Fokker-Planck equation describing the spatial evolution of the density $f_{E_p}(E_p, x)$ over the energy domain, namely,

$$\frac{\partial f_{E_p}(E_p, x)}{\partial x} = -\frac{\partial}{\partial E_p}[k_1(E_p, x) f_{E_p}(E_p, x)]$$

$$+ \frac{\partial^2}{\partial E_p^2}[k_2(E_p, x) f_{E_p}(E_p, x)]. \tag{18}$$

In this problem, unlike the examples above, the drift coefficient $k_1(E_p, x)$ [M L T$^{-2}$] has two parts: a fixed spatial rate $k_{1h} = mg\sin\theta$ of gravitational heating due to conversion of potential to kinetic energy, and a spatial cooling rate $k_{1c}(E_p, x) \approx mg\mu\cos\theta$ due to collisional friction. These are interpreted as average spatial rates of change in particle energy per unit distance; each is a force. That the drift coefficient involves two parts is similar to the description of particle motions in soils provided by





Furbish et al. (2009b), where gravitational settling motions are distinguished from scattering motions. The quantity $k_2(E_p, x)$ is a diffusion coefficient equal to one-half the spatial rate of change in the variance of particle energy states. Not shown in Eq. (18) is an energy loss term due to deposition (Furbish et al., 2020a).

With this description of the spatial evolution of the density $f_{E_p}(E_p, x)$ in place, it then becomes straightforward to recast Eq.

(18) into dimensionless form in order to define a characteristic cooling length $X_{cA}$ associated with collisional friction. (Note that the analysis does not actually involve solving the Fokker-Planck equation, Eq. (18).) This quantity is entirely analogous to the advective time scale associated with an advection (or advection-diffusion) equation where, instead of time, it refers to a distance. Namely, in the absence of gravitational heating, $X_{cA}$ is a characteristic distance over which thermal collapse by advective cooling occurs due to collisional friction. This allows us to associate the spatial deposition rate with the advective

cooling rate. When the cooling term is integrated over all particle energy states (see Furbish et al., 2020a), the resulting characteristic length scale $L_c$ then has the role of an $e$-folding length of deposition, which connects the energy balance to the particle mass balance. Namely, for $N(x)$ particles,

$$\frac{\mathrm{d}N(x)}{\mathrm{d}x} = -\frac{N(x)}{L_c} = -\frac{\gamma m g \mu \cos\theta}{\alpha E_a(x)} N(x), \tag{19}$$

with deposition length $L_c = \alpha E_a(x)/\gamma m g \mu \cos\theta$. Moreover, this formulation has an important probabilistic attribute. From

Furbish et al. (2020a):

> "Note that the formulation does not involve specifying a threshold energy for deposition... Whereas low energy particles are on average more likely to become disentrained than are high energy particles, a set of particles with precisely the same low energy will for probabilistic reasons not be disentrained simultaneously. Each particle experiences a unique set of conditions that disentrain it; and because of this uniqueness of conditions a particle with
>
> 20 energy below an arbitrarily assigned threshold can with finite probability be gravitationally reheated to a higher energy state. For given particle and surface roughness conditions, the formulation treats this aspect of disentrainment as a probabilistic process... [in relation] to the distribution of particle energy states and the probabilistically expected extraction of energy during collisions."

What might an alternative formulation of deposition look like? (Having approached the problem as above, we admit that a

description of this sort represents a straw person to criticize. Nonetheless, we also can admit that our early efforts involved thinking of alternatives, so this criticism is not blind.) Suppose we start with the assumption that the deposition rate is inversely proportional to the average particle kinetic energy $E_a$, namely, $\mathrm{d}N(x)/\mathrm{d}x = -\lambda N$ with rate constant $\lambda \sim 1/E_a$ [L$^{-1}$]. This might be motivated heuristically by the idea that the likelihood of deposition decreases with increasing particle energy as characterized by the average energy $E_a(x)$ at position $x$. That is, for given slope and roughness conditions, the motions of high

energy particles are less likely to be arrested during collisions than are those of low energy particles, and the average energy $E_a(x)$ is a measure of the overall energetic state of the particles. According to Eq. (19), this assumption actually would provide a very good start on the problem! (This assumes all elements of $\lambda = 1/L_c$ could be deduced.) However, this approach would still require a separate formulation of how the energy $E_a(x)$ varies with position $x$. And it would risk missing a critical step





revealed in the full analysis, that a key average is the harmonic average energy $E_h = E_a/\gamma$, which determines how deposition influences the energy balance. Moreover, the full energy balance is needed to specify the disentrainment rate as in Eq. (19), thence leading to the derivation of the generalized Pareto distribution of travel distances. We suggest that this example, details of which are provided in Furbish et al. (2020a), offers a clear view of the value of a statistical mechanics approach involving

the Fokker-Planck equation, highlighting the mechanistic yet probabilistic nature of the analysis.

### 4.1.3 The generalized Pareto distribution as a maximum entropy distribution

Our third example highlighting the purpose and merits of a probabilistic description of particle motions and transport is centered on demonstrating that the generalized Pareto distribution is a maximum entropy distribution. We again start with some background, briefly outlining the origin of the idea of a maximum entropy distribution.

The canonical example of a maximum entropy distribution is the Boltzmann distribution of the energy states $\epsilon$ of gas particles at thermal equilibrium. For a great number $N$ of gas particles with a fixed total energy $E$, the original derivation of the Boltzmann distribution involves enumerating the total number of accessible microstates — the many different ways to arrange $N$ particles into energy states where each arrangement possesses the same fixed total energy $E$ — then determining the most probable arrangement. (This idea is illustrated in Figure 3 in Furbish and Schmeeckle (2013).) Schrödinger (1946, p. 6)

succinctly describes the matter. Using his notation, consider the energy states $\epsilon_1$, $\epsilon_2$, $\epsilon_3$, …, $\epsilon_l$, …. Then let $a_1$, $a_2$, $a_3$, …, $a_l$, … denote the number of independent systems in each energy state. (Here it suffices to imagine that a 'system' consists of an individual particle.) We then imagine a set of macrostates ("classes") where each macrostate involves a set of $N$ systems arranged into energy states. Namely,

"all [micro]states of the assembly are embraced — without overlapping — by the classes [macrostates] de-
scribed by all different admissible sets of numbers $a_l$... The number of single [micro]states, belonging to this
class [macrostate], is obviously

$$P = \frac{N!}{a_1!a_2!a_3!\ldots a_l!\ldots}\,. \tag{20}$$

The set of numbers $a_l$ must, of course, comply with the conditions

$$\sum_l a_l = N\,, \quad \sum_l \epsilon_l a_l = E\,. \tag{21}$$

The present method [of the most probable distribution] admits that, on account of the enormous largeness of
the number $N$, the total number of distributions (i.e. the sum of all $P$'s) is very nearly exhausted by the sum of
those $P$'s whose number sets $a_l$ do not deviate appreciably from the set which gives $P$ its maximum value (among
those, of course, which comply with [Eq. (21)]). In other words, if we regard this set of occupation numbers as
obtaining always, we disregard only a very small fraction of all possible distributions — and this has 'a vanishing
likelihood of ever being realized'."

The procedure thus amounts to choosing the macrostate containing the greatest number of microstates, each consistent with fixed $N$ and $E$. This is achieved by applying Stirling's approximation to Eq. (20), taking the differential of the resulting





expression and of Eq. (21) with respect to $a_l$ and setting these to zero, then using Lagrange multipliers to determine that

$$a_l = N \frac{e^{-\lambda \epsilon_l}}{\sum_l e^{-\lambda \epsilon_l}} . \tag{22}$$

This is effectively the Boltzmann distribution. The Lagrange multiplier $\lambda = 1/k_B T$, where $k_B$ is the Boltzmann constant and $T$ is temperature, is determined independently. A key point of the derivation is that Stirling's approximation of Eq. (20) yields a

term representing the Gibbs entropy of the system. So in choosing the macrostate containing the greatest number of microstates, the procedure is equivalent to choosing the distribution whose entropy is a maximum relative to all other possible choices.

From Furbish et al. (2020c):

"Jaynes (1957a, 1957b) elaborated the significance of the fact that the Gibbs entropy in statistical mechanics and the Shannon entropy in information theory are essentially one and the same, differing only by a constant.

This similarity inspired Jaynes to champion the use of a maximum entropy criterion in choosing a probability distribution, leading to what is now known as the maximum entropy method... [W]hether viewed as a method of statistical mechanics or as one of inferential statistics... it provides an unbiased choice of a distribution by honoring only what is known mechanically about a system. That is, this unbiased choice is a maximally noncommittal choice that is faithful to what we do not know; it is therefore the most reasonable choice in the absence of additional

information..."

Within this context, there are three notable elements in our effort (Furbish et al., 2020c) to demonstrate that the generalized Pareto distribution as applied to the rarefied motion problem is a maximum entropy distribution. First, in this work we noted that "constraints imposed on the system normally translate to constraints imposed on the moments of the distribution.... [in which] case the method leads to a distribution that is among the exponential family (e.g., exponential, Gaussian). However,

applications of the maximum entropy method to non-exponential distributions, including heavy-tailed distributions, are of particular interest in many problems (Peterson et al., 2013)." Moreover, recall that the generalized Pareto distribution has three forms: it is bounded for shape parameter $A < 0$, heavy-tailed for $A > 0$ and exponential for $A = 0$ (Figure 1). If we are to suggest that the generalized Pareto distribution is for mechanical reasons a maximum entropy distribution, then it becomes essential to show that all three of its forms are constrained in the same manner — as opposed to appealing to separate constraints

for each of the three forms. Indeed, the energetic basis of the disentrainment rate, Eq. (4), provides this common constraint. It allows us to calculate an energetic "cost" — the total cumulative energy extracted by collisional friction per unit kinetic energy available during particle motions. This energetic cost is entirely analogous to that associated with the economics of scale (Peterson, 2013), where net heating contributes to an energetic "discount" that allows particles to achieve larger distance states $x$, and net cooling imposes a "penalty" that suppresses long distance motions. In effect "the analysis represents a novel

generalization of an energy-based constraint in using the maximum entropy method to infer non-exponential distributions — to include the versatile properties (forms) of the generalized Pareto distribution as applied to the rarefied particle motion problem." Stepping back, we suggest that similar considerations of particle energetics may be useful for clarifying the behavior of particles in other systems.



Second, the versatile form of the generalized Pareto distribution is rather unusual. Numerous well-known distributions of course take the form of a related distribution for certain parametric values. Nonetheless, the generalized Pareto distribution is distinctive in that it has a bounded form ($A < 0$) that decays faster than an exponential distribution with the triangular and uniform distributions as special cases, a heavy-tailed form ($A > 0$) with undefined mean or variance for sufficiently large values

of the shape parameter $A$, and an exponential form ($A = 0$) separating its bounded and heavy-tailed forms. Moreover, that this distribution seems to correctly describe the energetics of rarefied particle motions for varying slope and surface roughness conditions representing all three forms (Figure 2) is at first glance astonishing — notably including the abrupt mathematical transition between bounded and heavy-tailed forms. That the constraint provided by a fixed energetic cost relative to the available kinetic energy provides a unifying explanation of the three behaviors lends confidence that each of the three forms

of the distribution represents the most probable arrangement of distance states — just as the Boltzmann distribution represents the most probable arrangement of energy states of gas particles at thermal equilibrium. Nothing special or unusual changes in the physics of disentrainment in the transition from the bounded form to the heavy-tailed form of the distribution in crossing isothermal conditions — a point of clarity provided by the maximum entropy analysis.

This result also adds clarity to the idea of nonlocal versus local transport (Metzler and Klafter, 2000; Schumer et al., 2009;

Foufoula-Georgiou et al., 2010; Furbish and Roering, 2013). In studies of tracer particle transport, and setting aside the effects of particle rest times, local behavior is associated with a light-tailed distribution of particle displacements during a small interval $\mathrm{d}t$, leading to Gaussian dispersion. Nonlocal behavior is associated with a heavy-tailed distribution of displacements leading to non-Gaussian (anomalous) dispersion as represented by, say, a fractional advection-diffusion equation with respect to space (Metzler and Klafter, 2000; Schumer et al., 2009). In comparison, consider the situation in which the shape parameter

$A$ of the generalized Pareto distribution is small, positive or negative. With small $A < 0$ the light-tailed form of the distribution has an upper bound at $x = B/|A|$ (Figure 1), and as $A$ approaches zero from below, this upper bound may become exceedingly large but nonetheless remains finite. With small positive $A > 0$ close to zero the distribution has "flipped" to an unbounded heavy-tailed form (with finite mean and variance). Yet any difference in the physics of particle motions up to $x = B/|A|$ for $A$ positive or negative (and small) is imperceptible. Indeed, except near the upper bound $x = B/|A|$ and beyond, the two forms of

the distribution for $A$ close to zero are essentially indistinguishable — the difference representing a mathematical precision far beyond what one might be capable of detecting from measurements of particle travel distances (Furbish et al., 2020b). Thus, within the context of tracer particle behavior, whereas local versus nonlocal behavior defined in terms of the distribution form may be an important mathematical distinction — notably if the distribution involves undefined moments — this distinction offers limited insight regarding the mechanical interpretation of particle motions. Hence, in the problem of rarefied particle

motions on hillslopes, and consistent with physical interpretations of nonlocal behavior (Bocquet et al., 2009; Brantov and Bychenkov, 2013; Henann and Kamrin, 2013), the idea of nonlocal transport as embodied in Eq. (1) and Eq. (2) reminds us that the flux or its divergence at position $x$ is determined by factors influencing entrainment and particle motions "far" upslope from this position (Furbish and Roering, 2013; Furbish et al., 2016, 2020a; Doane, 2018; Doane et al., 2018).

Third, focusing on the second part of the quotation above, the maximum entropy method reminds us of the value of the

principle of parsimony — appealing to the simplest explanation consistent with available evidence — in the presence of uncer-





tainty. Boltzmann did not know *a priori* the distribution of gas particle energy states, Eq. (22); he imposed only the constraints of a fixed number of particles and a fixed total energy. The maximum entropy derivation thus honored his understanding of the system, but no more. In effect the derived distribution of energy states — including the foundational assumption that each accessible microstate is equally probable — became an hypothesis to be tested against experimental observations (Tolman,

1938). With respect to applications of the maximum entropy method to sediment particle motions, we "highlight the fact that a distribution thus chosen is not necessarily the "correct" distribution (Furbish et al., 2016)... It is the most reasonable choice in the absence of additional information... [and in] this sense the maximum entropy method is a formal application of Occam's razor" (Furbish et al., 2020c). We therefore suggest that this represents one viable element of a strategy to deepen our mechanical understanding of attributes of particle motions that we observe, measure and describe statistically. As noted by Ancey (2020b)

in relation to bed load transport, "One strength of entropy-based methods is their use of the physical information conveyed by data, thereby enforcing physical consistency... [opening] new avenues of research combining statistical information and physics-based models." On this point we note that a distribution selected according to a maximum entropy criterion may serve as an ideal prior hypothesis in subsequent analysis, including Bayesian analysis (Jaynes, 1988).

## 4.2 Rarefied versus continuum conditions

### 4.2.1 Motivation

Perhaps it is obvious that in this problem a description of the physics of particle motions cannot meaningfully start from the idea of continuum behavior. Particle motions are patchy and highly intermittent, and in most settings these motions are far from conditions that could be considered continuum-like granular flows. Particle behavior is dominated by particle-surface interactions rather than particle-particle interactions, and the conventional idea of appealing to a Knudsen number to ascertain

continuum behavior is irrelevant. Moreover, aside from descriptions of the physics of particle motions, in the absence of continuum conditions we cannot justifiably appeal to familiar continuum-like definitions of the particle flux and its divergence that are based on the assumption that particle number densities and locally averaged velocities are well defined quantities that vary smoothly over space and time. Yet the particle flux and its divergence are of particular interest in many problems, and we therefore turn our focus to a thorough description of these quantities.

As noted above, precise definitions of the sediment particle flux and its divergence do not assume continuum conditions at the outset (Ancey, 2010; Furbish et al., 2012, 2016, 2017; Ancey and Pascal, 2020). For rarefied conditions these definitions are translated into probabilistic expressions, of which the entrainment forms of the flux and the Exner equation, Eq. (1) and Eq. (2), are examples. Of concern, then, is the meaning and use of continuously differentiable functions in these nonlocal expressions, namely, the entrainment rate $E_s(x,t)$, the probability density $f_r(r;x,t)$ of travel distances $r$ and the associated

exceedance probability function $R_r(r;x,t)$. At risk is misinterpreting these continuous probabilistic functions as implying a continuum-like description of transport such that the flux $q(x,t)$, if defined as a time-averaged quantity, may be envisioned as varying continuously in space and time akin to continuum-like expressions of transport — the canonical examples being the





process-response models introduced by Kirkby (1971) and Carson and Kirkby (1972) involving expressions of the flux that vary with surface configuration and semi-empirical rate constants. Similar comments pertain to the rate of change $\dot{\zeta}(x,t)$.

Consider the nonlocal expressions, Eq. (1) and Eq. (2). For simplicity of illustration we focus on a single particle size and rewrite these expressions as follows. Let $q_n(x,t)$ [L$^{-1}$ T$^{-1}$] denote the particle number flux at position $x$ and time $t$. Dividing Eq. (1) by the particle volume $V_p$ then gives:

$$q_n(x,t) = \int_{-\infty}^{x} E_n(x',t) R_r(x - x'; x',t) \, \mathrm{d}x',$$ (23)

where $E_n(x,t)$ [L$^{-2}$ T$^{-1}$] denotes the particle number entrainment rate. In turn, let $n(x,t)$ [L$^{-2}$] denote the areal particle number density. Dividing Eq. (2) by $V_p$ then gives:

$$\dot{n}(x,t) = \frac{\partial n(x,t)}{\partial t}$$

$$= -E_n(x,t) + \int_{-\infty}^{x} E_n(x',t) f_r(x - x'; x',t) \, \mathrm{d}x'.$$ (24)

In these expressions the flux $q_n(x,t)$, the number density $n(x,t)$ and the entrainment rate $E_n(x,t)$ are treated as continuous functions of position $x$ and time $t$. Moreover, the density $f_r(r; x,t)$ and the exceedance probability $R_r(r; x,t)$ are continuous functions of the travel distance $r$, and the forms of these functions are considered to vary smoothly with $x$ and $t$. However, Eq. (23) and Eq. (24) do *not* imply that transport may be envisioned as a continuum-like behavior or that the flux and its divergence vary smoothly with position $x$ and time $t$ in any particular setting. Rather, these are probabilistic expressions that represent ensemble expected conditions, not the outcome of any individual realization (Appendix C). In fact, both the flux $q_n(x,t)$ and the rate $\dot{n}(x,t) = \partial n(x,t)/\partial t$ are to be considered random variables due to rarefied transport conditions.

To illustrate these points, here we consider an idealized situation in which the entrainment rate $E_n(x,t)$ is Poisson in time and space, but modified to include effects of intermittency and patchiness. The cases presented next, although involving approximations of the entrainment process, nonetheless suffice to illustrate the consequences of a noise driven process. This includes an explicit definition of ensemble expected conditions versus the outcome of an individual realization, and the relation between these and time-averaged conditions. The presentation reflects elements of the analysis of Ancey and Pascal (2020) concerning bed load transport.

### 4.2.2 Line source

The idea of a line source of sediment particles delivered to a hillslope is embodied in the experimental and field-based work of Kirkby and Statham (1975) and Statham (1976) concerning the motions and downslope sorting of particles on scree slopes. Here we consider a simple version of this problem.

**Flux with Poisson delivery rate:** We start by envisioning a planar hillslope at the base of a cliff. Particles are delivered from the cliff to the top of the hillslope as a line source at $x = 0$, and we neglect particle entrainment on the hillslope. In this





situation we may simplify the notation. Namely, the travel distance $r \rightarrow x$ so the density $f_r(r;x) \rightarrow f_x(x)$ and the exceedance probability $R_r(r;x) \rightarrow R_x(x)$. In turn, the entrainment rate $E_n$ is reinterpreted as a boundary flux, the expected number $n$ of particles delivered to the top of the hillslope per unit width per unit time. For a width $\Delta y$, the expected number of particles delivered per unit time is $\eta = E_n \Delta y$. The number $n$ of particles delivered during an interval $\tau$ is then given by the Poisson

distribution, namely,

$$f_n(n, \tau, x = 0) = \frac{(\eta \tau)^n}{n!} e^{-\eta \tau}, \tag{25}$$

with mean $\mu_n = \eta \tau$ and variance $\sigma_n^2 = \eta \tau$. More generally, the expected number of particles reaching position $x$ per unit time is $\eta R_x(x)$, and the number $n$ of particles reaching $x$ during $\tau$ is again given by a Poisson distribution,

$$f_n(n, \tau, x) = \frac{[R_x(x) \eta \tau]^n}{n!} e^{-R_x(x) \eta \tau}, \tag{26}$$

with mean $\mu_n = R_x(x) \eta \tau$ and variance $\sigma_n^2 = R_x(x) \eta \tau$. For reference below we note that the distribution of waiting times $w$ between successive events is exponential with mean $\mu_w = 1/\eta$ in the case of Eq. (25) and $\mu_w = 1/R_x(x)\eta$ in the case of Eq. (26). (Note that the waiting time $w$ should not be confused with a particle rest time.)

These results illustrate that the number $n$ of particles reaching $x$ during $\tau$ involves a distribution of possible outcomes whose variance increases linearly with the elapsed time $\tau$. For plotting we normalize this number by the expected number reaching $x$,

namely, $\hat{n}(x) = n(x)/\eta \Delta t$ with $\Delta t = 1$ (Figure 3). Note that the numbers used to generate these plots are somewhat arbitrary, as the width $\Delta y$ is not explicitly specified. That is, for given $E_n$, the numbers (and thus $\eta$) increase with $\Delta y$.

These plots may be interpreted several ways. First, for a specified delivery rate $\eta$ at the line source ($x = 0$), the plots (**a**) to (**d**) depict a decreasing exceedance probability $R_x(x)$ representing an increasing distance from the source for a fixed form of the distribution $f_x(x)$ of travel distances $x$. Second, the plots (**a**) to (**d**) depict a decreasing delivery rate $\eta$ at the source

viewed at $x = 0$. In this situation, downslope locations would display an increasing variability relative to the source conditions. For example, if Figure 3a represents conditions at the source ($x = 0$, $R_x(0) = 1$), then Figure 3c represents conditions at a downslope position $x$ with $R_x(x) = 0.01$. If Figure 3c represents conditions at the source, then Figure 3d represents conditions at a position $x$ with $R_x(x) = 0.1$. And, Figure 3a could represent conditions at a position $x \gg 0$ with $R_x(x) = 0.01$ for a rate $\eta$ that is $10^2$ larger. Third, if in Figure 3a the rate $\eta = 1000$ represents the expected number of events per year, then the signals in

Figure 3d could represent this same rate, but plotted in terms of expected events per "milli-year" (or one event per 8.8 hours). Note that individual realizations never converge to ensemble expected values, although the relative variability decreases with large $\eta$. That is, the coefficient of variation decreases slowly as $[R_x(x)\eta \tau]^{-1/2}$, but this is not a mean-reverting process.

The plots in Figure 3 also may be reinterpreted in terms of variations in the form of the generalized Pareto distribution with respect to hillslope positions $x$. That is, each plot may represent different positions of $x$ coinciding with the same exceedance

probability $R_x(x)$ for different values of the shape and scale parameters $A$ and $B$. For example, assuming fixed $B$, each plot may represent a relatively small value $x > 0$ with small $A$ and a relatively large $x > 0$ with large $A$. Or, for a fixed position $x > 0$, Figure 3c might represent a distribution with small $R_x(x)$ coinciding with small $A$ whereas Figure 3a might represent a distribution with large $R_x(x)$ coinciding with large $A$. In general the position $x$ associated with a specific value of $R_x(x)$ increases with $A$ for given $B$. Note that no particles move past a position $x$ beyond the upper bound $B/|A|$ for $A < 0$.



Earth **Surface**
**Dynamics**
Discussions



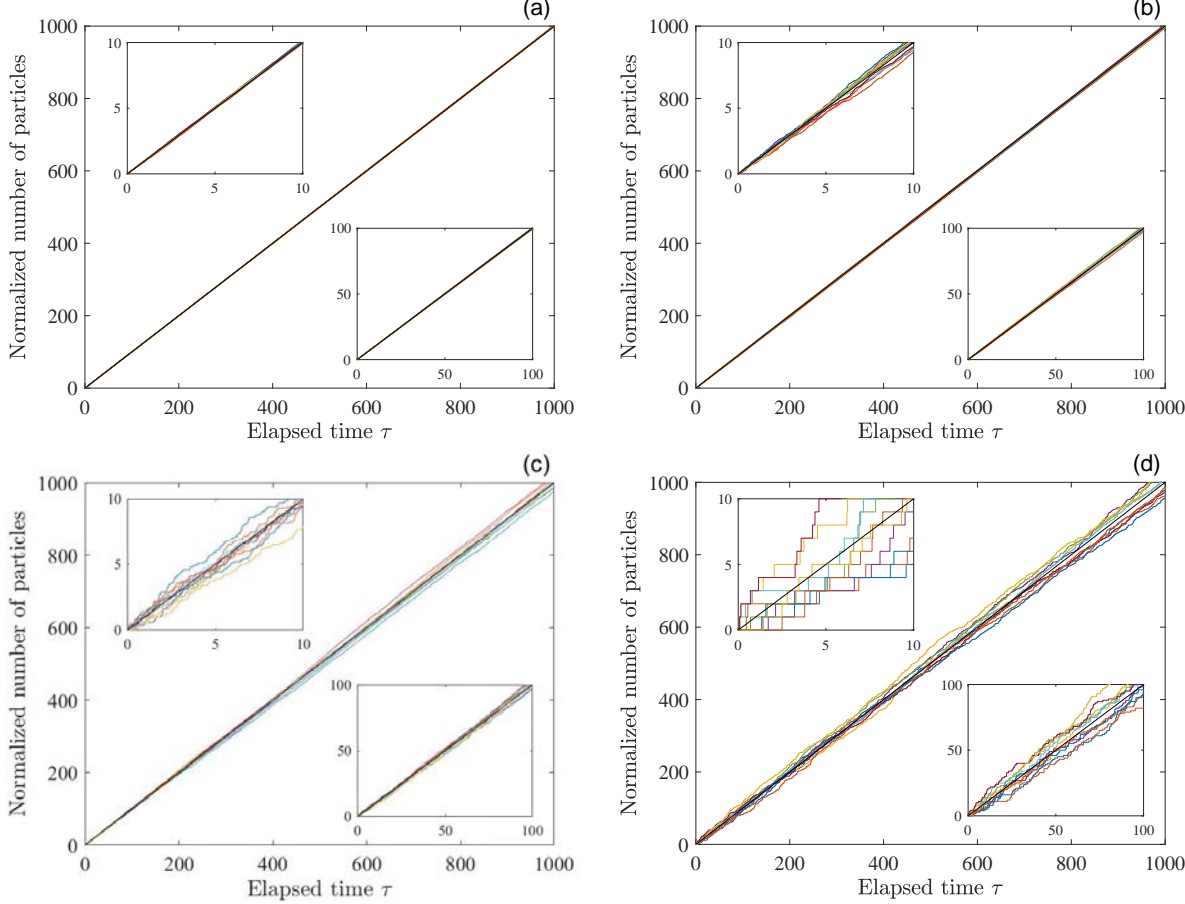

**Figure 3.** Plot of ten realizations (colored lines) of normalized number $\hat{n}(x)$ versus elapsed time $\tau$ showing increasing variance with $\tau$. Plots generated with (**a**) $\eta = 1000$, (**b**) $\eta = 100$, (**c**) $\eta = 10$ and (**d**) $\eta = 1$. Black line represents ensemble expected values of $\hat{n}(x)$. Compare with Figure 2 in Ancey and Pascal (2020).

In turn, we compute the normalized time-averaged particle number flux as $\hat{q}_n(x) = \hat{n}(x)/\tau$ (Figure 4). This flux also is a random variable. It exhibits relatively large variability with small elapsed times $\tau$, then converges to the ensemble expected value with increasing $\tau$. The rate of convergence decreases with decreasing delivery rate $\eta$, or with decreasing exceedance probability $R_x(x)$ representing an increasing distance from the line source.

5    Ancey and Pascal (2020) examine the more general question of estimating the time-averaged flux associated with a Poisson process (compare their Figure 2 with Figure 3). Within the context of measurements of bed load sediment transport, they show how the variability in estimates of the time-averaged flux vary with the measurement interval, and present a re-sampling (bootstrap) protocol for assessing how the variance of the flux varies with the sampling interval based on an individual realization. As noted below, however, we rarely if ever have time series needed to support this type of analysis when describing slow 10   systems.





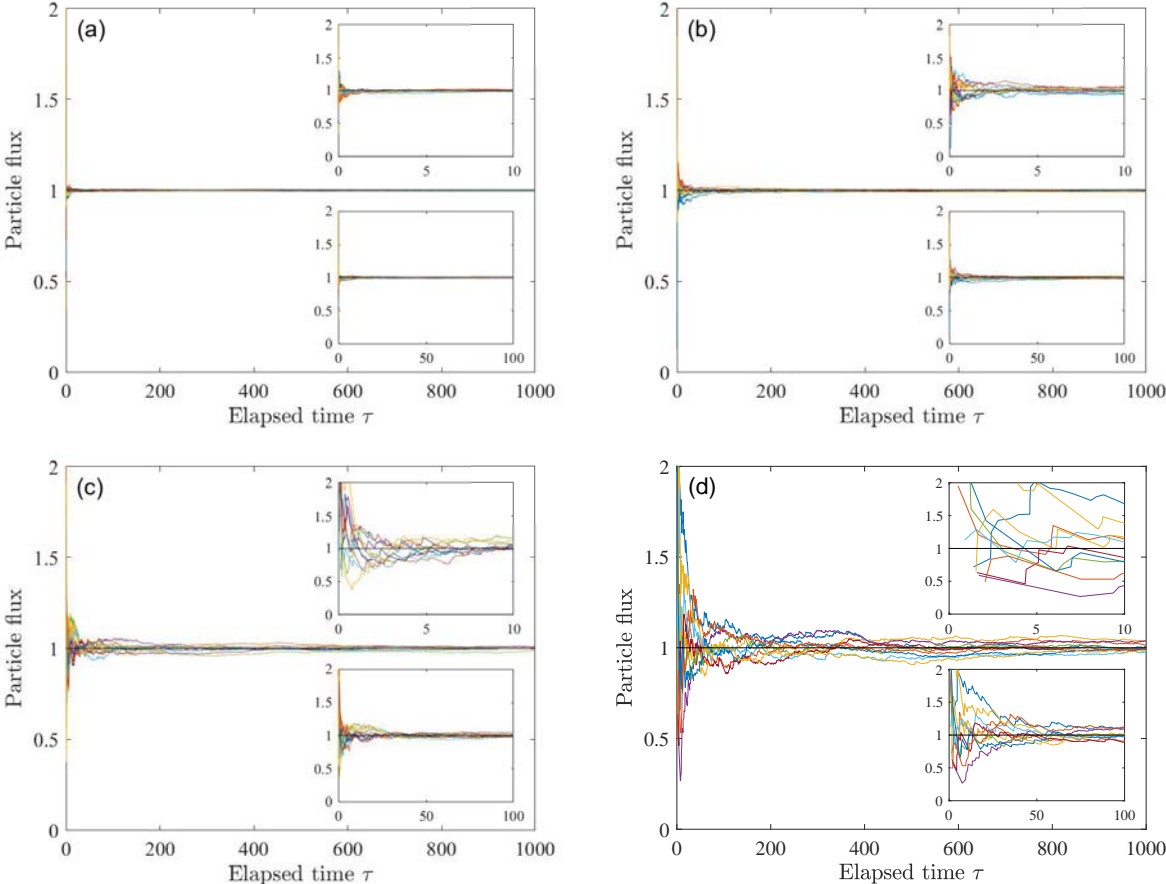

**Figure 4.** Plot of ten realizations (colored lines) of normalized time-averaged particle number flux $\hat{q}_n(x)$ versus elapsed time $\tau$ showing convergence to ensemble expected value (black line) with increasing $\tau$. Plots coincide with conditions in Figure 3. Note that initial values start at $\tau > 0$.

With this example in place we offer an explicit definition of the idea of ensemble expected conditions for a Poisson process. The word 'ensemble' refers to a great number $N_e$ of nominally identical but independent systems, each subject to the same physics of random events characterized by the rate constant $\eta$ (Appendix D) for specific values of $A$ and $B$. In this view, each realization plotted in Figure 3 represents the outcome of one system in the ensemble. (Note that this meaning of 'ensemble' is quite different from the oft used description of an ensemble of particles.) For any individual system there is one possible outcome $n$ at time $\tau$ with probability given by Eq. (25) or Eq. (26). For an ensemble of systems all possible outcomes $n = 0, 1, 2, 3, \ldots$ exist at time $\tau$ in the proportions given by Eq. (25) or Eq. (26). Thus, ensemble conditions in this example refer to the distribution of possible outcomes with a well-defined ensemble average and variance. In turn, the particle flux $q_n(x,t)$ given by Eq. (23) represents the ensemble expected flux, not the flux associated with any particular realization. Similarly, as



described further below the rate of change in particle numbers $\dot{n}(x, t)$ given by Eq. (24) represents the outcome defined by ensemble expected conditions, not the rate associated with any realization.

**Flux with intermittent delivery rate:** The idea of a Poisson delivery rate nicely illustrates the growing variance of particle numbers associated with a simple noise driven process. However, of particular interest are effects of an intermittent delivery rate

— recognizing that this rate likely involves fluctuations in particle numbers with seasonal to longer-term variations in factors influencing particle entrainment. Again consider a line source of particles at $x = 0$. We assume that events are Poisson with an expected rate $\eta$, where each event, rather than representing one particle, instead involves $n$ particles described by a specified distribution. Results described below are qualitatively insensitive to the choice of this distribution, so for simplicity we use an (integer) exponential distribution with mean $\mu_n$, which provides skew in the number of particles per event. In this situation the

delivery of particles at $x = 0$ is no longer a Poisson process; increasing intermittency is represented by a decreasing rate $\eta$.

As a point of reference, Benjamin et al. (2020) provide an assessment of efforts to observe and measure rockfall events contributing to cliff erosion and thus to downslope delivery of particles. The frequency and magnitude of these events may vary widely, from the chronic activity of small rockfall events to large infrequent events, depending on the geological and environmental factors that influence the mechanisms of weathering and failure (Luckman, 2013; Strunden et al., 2015; Mair

et al., 2020). The frequency of occurrence of rockfall volume typically varies as an approximate inverse power function of volume, where the specific relation depends on the spatial coverage and temporal duration of the data set (Benjamin et al., 2020). Rockfall volumes do not translate directly to particle numbers, both of which are influenced by the geometry of cliff rock fracturing and fragmentation (Domokos et al., 2020; Verdian et al., 2020), and impact shattering (Luckman, 2013). Nonetheless, these observations point to the inherent stochasticity of rockfall over many scales, including variations in intermittency with

time (Section 4.3.2).

Like a Poisson particle delivery rate (Figure 3), the number of particles $n$ reaching $x$ during $\tau$ involves a distribution of possible outcomes whose variance increases with the elapsed time $\tau$ (Figure 5). Increasing the average number of particles per event, $\mu_n$, tends to decrease the variability about the ensemble expected values. However, this variability is far more influenced by the degree of intermittency. Namely, the variability increases with decreasing $\eta$, and is notably larger than that associated

with a purely Poisson process (Figure 3). For comparison, each of the plots in Figure 5 involve 100 times fewer events per year than the corresponding plot in Figure 3, but 10 times the number of expected particles per year. The plots may be interpreted in a manner similar to that given above for Figure 3. Although not shown to save space, the normalized time-averaged particle flux $\hat{q}_n(x)$ is similar in appearance to that associated with a Poisson delivery rate (Figure 4), but with larger variability and slower convergence to the ensemble expected value with increasing $\tau$.

**Steady deposition:** Consider the rate $\dot{n}(x) = \partial n(x)/\partial t$ in Eq. (24). In the idealized situation involving a line source of particles arriving at $x = 0$ with rate $\eta$, and in the absence of particle entrainment, the expected rate of deposition within the small interval $\Delta x$ at position $x$ is $\dot{n}(x) \approx f_x(x)\Delta x \eta$. However, for an individual realization this rate is discontinuous in time. Let $\Delta n$ denote the number of particles deposited within a specified interval $\Delta t$. The time-averaged rate of deposition during $\Delta t$





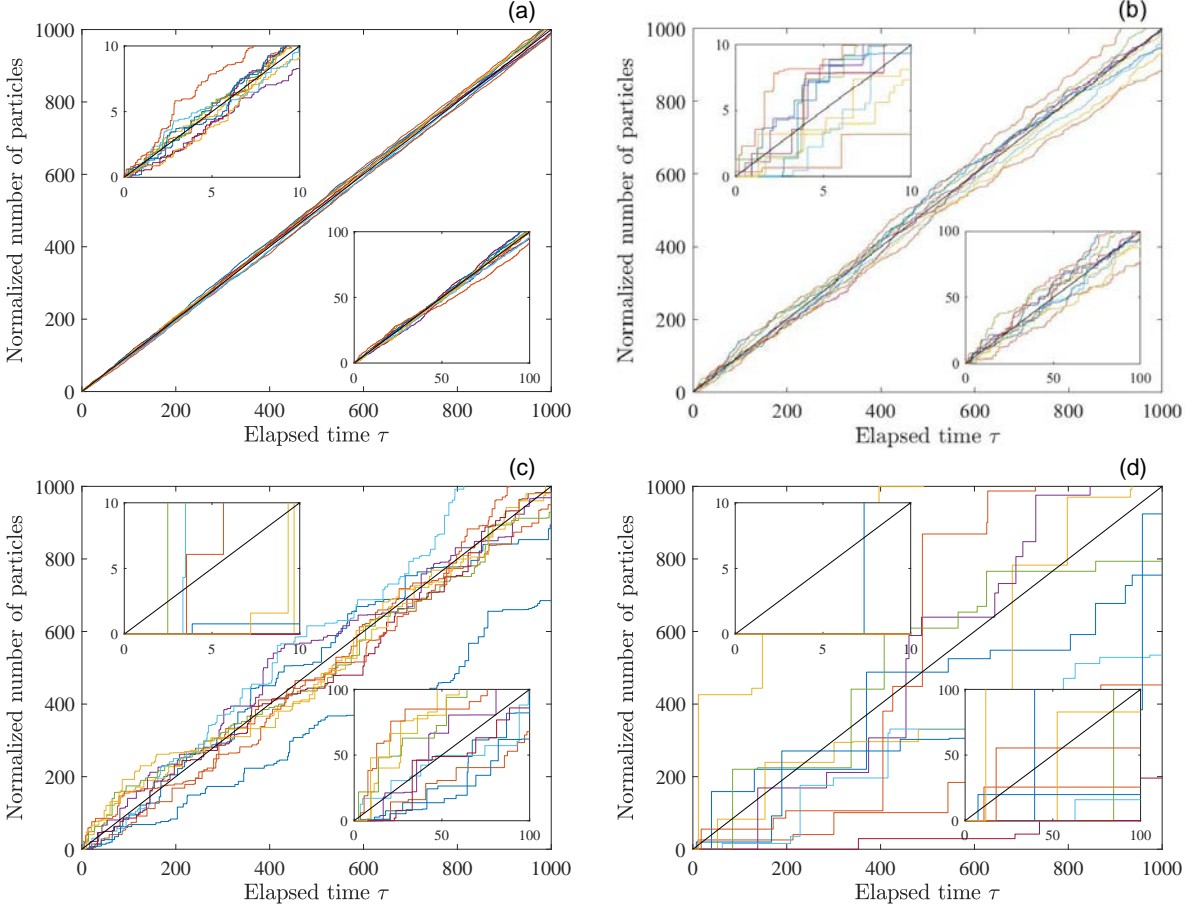

**Figure 5.** Plot of ten realizations of normalized number $\hat{n}(x)$ versus elapsed time $\tau$ showing increasing variance with $\tau$. Plots generated with $\mu_n = 1000$ and (**a**) $\eta = 10$, (**b**) $\eta = 1$ (**c**) $\eta = 0.1$ and (**d**) $\eta = 0.01$. Red line represents ensemble expected values of $\hat{n}(x)$.

is then $\Delta n/\Delta t$. The number of particles deposited with $\Delta x$ at $x$ during $\Delta t$ is then again distributed as a Poisson distribution,

$$f_{\Delta n}(\Delta n, \Delta t, x) = \frac{[f_x(x)\Delta x \eta \Delta t]^{\Delta n}}{(\Delta n)!} e^{-f_x(x)\Delta x \eta \Delta t}, \tag{27}$$

with mean $\mu_{\Delta n} = f_x(x)\Delta x \eta \Delta t$ and variance $\sigma^2_{\Delta n} = f_x(x)\Delta x \eta \Delta t$. Thus, as above, the variance of the numerator of $\Delta n/\Delta t$ increases indefinitely with the interval $\Delta t$.

5      For a Poisson process, the deposition events $\Delta n$ within any successive interval $\Delta t$ are independent. Letting $k = 1, 2, 3, ...$ denote successive intervals, then $\tau = k\Delta t$. This means that summing the number of particles deposited during successive intervals $\Delta t$ is the same as summing over the total elapsed time $\tau$. Thus, deposition at $x$ proceeds as a random process whose appearance is qualitatively similar to the examples above (Figure 3). The rate $\Delta n/\Delta t$ similarly converges slowly to the ensemble value $\dot{n}$ with increasing time interval $\tau$. Similar conclusions apply to the case of an intermittent delivery rate of



particles. The expected deposition rate $\dot{n}(x)$ thus represents the ensemble average, not the rate of individual realizations except in the limit of $\tau = k\Delta t \to \infty$.

### 4.2.3 Distributed entrainment

The idea of distributed entrainment is embodied in the work of Doane (2018) and Doane et al. (2018) concerning nonlocal
sediment transport. This work involves numerical simulations of the time evolution of the profiles of steep lateral moraines in the Sierra Nevada, California, for comparison with field-based measurements. It examines entrainment that occurs over the entire moraine profile due to disturbances and the role of vegetation in sediment capacitance — the capture, storage and release of sediment (Furbish et al., 2009a; Lamb et al., 2011, 2013; DiBiase and Lamb, 2013; Doane, 2018; Doane et al., 2018). Here we consider a simple version of this problem involving uniformly random entrainment.

**Flux with Poisson entrainment:** In contrast to a line source, here we envision a uniformly random entrainment rate $E_n$ over the domain $x$. We return to our original notation involving travel distances $r$ with probability density $f_r(r;x)$ and exceedance probability $R_r(r;x)$, neglecting variations with time $t$.

For a uniformly random entrainment rate $E_n(x) = E_n$ the expected number of particles reaching $x$ per unit time is

$$\eta = E_n \int\limits_{-\infty}^{x} R_r(x - x'; x') \, \mathrm{d}x' \,. \tag{28}$$

The integral in Eq. (28) is equal to the mean travel distance $\mu_r$, if this moment exits. Thus, $\eta = E_n \mu_r$, which is identical to the definition of the particle flux provided by Einstein (1950) for steady uniform bed load transport. The expected number $n$ of particles reaching position $x$ during $\tau$ is $\eta\tau$, so

$$f_n(n, \tau, x) = \frac{(\eta\tau)^n}{n!} e^{-\eta\tau} \,, \tag{29}$$

with mean $\mu_n = \eta\tau$ and variance $\sigma_n^2 = \eta\tau$. Note that Eq. (29) is identical in form to Eq. (25) and Eq. (26). This means that
the number of particles reaching position $x$ varies in a manner that is qualitatively the same as depicted in Figure 3, and the time-averaged flux is qualitatively the same as depicted in Figure 4. For a given mean travel distance $\mu_r$, convergence to the expected value $\eta = E_n \mu_r$ therefore strongly depends on the entrainment rate $E_n$. This convergence decreases with increasingly rarefied conditions.

If the integral in Eq. (28) does not converge — that is, the mean travel distance is undefined — then the expected particle
flux is undefined. This coincides with a shape parameter $A \geq 1$ for the generalized Pareto distribution. This also means that the expected divergence of the flux (see below) is undefined. However, we must be cautious to not over-interpret this result, as Eq. (28) assumes the upslope integration of the exceedance probability $R_r(r;x)$ is unbounded. In reality, the integration associated with any position $x$ extends only to the hillslope crest, thus truncating $R_r(r;x)$ such that the integral in Eq. (28) is finite. Nonetheless, numerical simulations confirm that the expected flux increases indefinitely as the upslope distance of integration
increases. The implication is this: If the distribution of particle travel distances is heavy-tailed with undefined mean, and if the heavy-tailed distribution applies to much or all of the hillslope, then the expected flux at the base of the hillslope depends on





its length. Indeed, if the mean travel distance is undefined and disentrainment is negligible (i.e., few if any particles stop), then the expected flux at the base of the hillslope is essentially equal to the entrainment rate integrated over the entire hillslope. This represents the "entrainment limited" analogue of "detachment limited" conditions. Furbish et al. (2020b) report estimates of $A \geq 1$ for several of the field-based experiments reported by DiBiase et al. (2017) and Roth et al. (2020).

**Flux with patchy entrainment:** Consider uniformly random entrainment events, where each event, rather than representing one particle, instead involves $n$ particles. As above we use an (integer) exponential distribution with mean $\mu_n$. In this situation the random location of events involving different numbers of particles yields a spatial patchiness in particle entrainment. If $E_n$ now denotes the event rate, then the expected rate at which particles reach position $x$ is $\eta = E_n \mu_n \mu_r$. However, unlike the previous example, this is not a Poisson process. Nonetheless, similar to the conclusions above, numerical simulations reveal that the number of particles reaching position $x$ varies in a manner that is qualitatively the same as depicted in Figure 3, and the time-averaged flux is qualitatively the same as depicted in Figure 4. For a given mean travel distance $\mu_r$ and mean number of particles $\mu_n$, convergence to the expected value $\eta = E_n \mu_n \mu_r$ therefore strongly depends on the rate $E_n$. This convergence decreases with increasingly rarefied conditions.

As in the preceding example, if the integral in Eq. (28) does not converge — that is, the mean travel distance is undefined
— then the expected particle flux is undefined. The implications of this are similar to those described above regarding Poisson entrainment.

**Divergence with Poisson entrainment:** Consider the changing number $n(t)$ of particles within an element $\Delta x$ as particles arriving from upslope are deposited and particles entrained within the element move downslope. Assuming that uniformly random entrainment is a Poisson process with expected entrainment rate $E_n$, then the expected particle number flux into the
element $\Delta x$ is equal to the expected flux out of this element. Specifically, the rate at which particles moving into $\Delta x$ become deposited within this element is

$$E_n \int\limits_{-\infty}^{x} R_r(x - x'; x')\, \mathrm{d}x'$$

$$-E_n \int\limits_{-\infty}^{x} R_r(x + \Delta x - x'; x')\, \mathrm{d}x'. \tag{30}$$

The first term in Eq. (30) is the rate at which particles reach $x$ from upslope. The second term is the rate at which particles reaching $x$ from upslope move past the interval $\Delta x$. The rate at which particles are entrained within $\Delta x$ and leave this element is

$$-E_n \int\limits_{x}^{x+\Delta x} R_r(x + \Delta x - x'; x')\, \mathrm{d}x'. \tag{31}$$

Adding Eq. (30) and Eq. (31) then leads to the conclusion that, per unit width, the immigration rate $\eta_I$ is equal to the emigration
rate $\eta_E$. That is, $\eta_I = \eta_E = E_n \mu_r$, if the mean travel distance $\mu_r$ is defined.





Alternatively, for positions $x''$ such that $x \leq x'' \leq x + \Delta x$ the expected deposition rate $D_n$ within $\Delta x$, including particles that are entrained within this interval, is

$$E_n \int\limits_{x}^{x+\Delta x} \int\limits_{-\infty}^{x''} f_r(x'' - x'; x')\, \mathrm{d}x'\, \mathrm{d}x''. \tag{32}$$

The negative rate at which particles are entrained within $\Delta x$, including those that remain within this interval and those that
leave it, is

$$-E_n \int\limits_{x}^{x+\Delta x} \int\limits_{0}^{\infty} f_r(r; x'')\, \mathrm{d}r\, \mathrm{d}x''. \tag{33}$$

The integrals involving $x'$ and $r$ are equal to unity, so upon summing Eq. (32) and Eq. (33) the expected rate $\dot{n} = -E_n + D_n = 0$ with $D_n = E_n$. That is, the local expected rate of deposition is equal to the expected rate of entrainment.

One might anticipate that the balance between expected immigration and emigration rates involving a uniform expected flux,
or between the expected deposition and entrainment rates, yields a particle number $n(t)$ within the element $\Delta x$ that fluctuates with time due to the randomness of the process, but which nonetheless is centered on a fixed mean value — a decidedly deterministic (continuum) point of view. This anticipated result, however, is incorrect. In fact, the number of particles within the element does not possess a stable distribution, and $n(t)$ is not a mean-reverting process. Instead, the number $n(t)$ undergoes an uncorrelated random walk over the $n$ domain, and with finite probability it may "wander" to an arbitrarily large or small
value. If the condition that $n(t)$ cannot become negative is imposed, then this is the well-known M/M/1 queuing problem (Stewart, 2009).

With $n(t) = 0, 1, 2, 3, ...$ and $\eta_I < \eta_E$, the stationary ensemble distribution $f_n(n)$ of states $n$ is a geometric distribution (the discrete version of an exponential distribution) with mean $\rho/(1 - \rho)$ and variance $\rho/(1 - \rho)^2$ where $\rho = \eta_I/\eta_E$. In contrast, with $\eta_I > \eta_E$ individual realizations $n(t)$ increase indefinitely, similar to the example of steady deposition above. Note that this
problem is just the beginning of a rich theory of dynamical systems falling under the headings of queuing theory, birth-death processes, Markov processes and generalized elastic models. We mention specific cases below, but otherwise the example above suffices to illustrate a basic, perhaps counterintuitive, outcome of noise driven processes.

There is little evidence that numbers $n(t)$ (i.e., local elevations $\zeta(t)$) on natural hillslopes exhibit unbounded (random-walk) behavior as in the example above. Entrainment and deposition do not lead to arbitrarily rough surfaces absent spatial correlation
in surface elevation. This reflects that additional physics becomes involved in the entrainment and deposition processes. Before addressing this point, first consider a related problem involving bed load transport, initially focusing on the number $n_a(t)$ of moving particles rather than the state of the streambed. The Markov birth-death formulation provided by Ancey et al. (2008) for rarefied transport conditions posits that if both the expected deposition rate and the expected emigration rate associated with an interval $\Delta x$ depend on the system state — the number $n_a(t)$ — then $n_a$ possesses a stable (ensemble) binomial distribution
$f_{n_a}(n_a)$ with mean $\mu_{n_a}$. If in addition collective entrainment proportional to $n_a(t)$ is involved (Ancey et al., 2008; Heyman, 2014; Heyman et al., 2014; Lee and Jerolmack, 2018; Pierce and Hassan, 2020), then $n_a$ is described by a negative binomial





distribution with well-defined mean and variance. The key lesson is this. With this noise driven process the distribution $f_{n_a}(n_a)$ of states $n_a$ is as important as the expected state $\mu_{n_a}$ in characterizing the process. Moreover, whereas the value of this expected state can be specified in terms of the rate constants associated with immigration, deposition and entrainment, the expected state has no more mechanical significance than other values of $n_a$; the expected (or modal) value is just more probable than other

values. With regard to the state $n(t)$ of the bed, because both deposition and collective entrainment depend on $n_a(t)$, these have a counterbalancing effect. In addition, Pierce and Hassan (2020) modify the formulation of Ancey et al. (2008) to couple erosion and deposition with the bed state $n(t)$. This explicitly includes a stabilizing feedback where entrainment preferentially occurs with aggradation and deposition preferentially occurs with degradation (Sawai, 1987; Wong et al., 2007). This coupling ensures that the state $n$ possesses a stable distribution with finite mean and variance.

Returning to transport on hillslopes, stabilizing effects may involve local changes in the entrainment rate due to effects of unstable particle configurations, collective entrainment of surface particles by moving particles, and preferential 'capture' of moving particles within local low spots or by roughness elements (Furbish et al., 2020b; Roth et al., 2020). Note that the rules of deposition in the particle-based transport model of Tucker and Bradley (2010) inherently provide this sort of stabilizing effect. Moreover, although attention has been given to the role of vegetation in sediment capacitance (Furbish et al., 2010; Lamb

et al., 2011, 2013; DiBiase and Lamb, 2013; Doane, 2018; Doane et al., 2018), this topic otherwise is largely unexplored in relation to modulating rates of entrainment and deposition over long time scales. In addition we may imagine a configuration involving (in a Fourier sense) a small-amplitude sinusoidal variation in surface elevation or roughness. The effect of this — including preferential deposition at certain locations — almost certainly would influence the behavior of particles whose motions start upslope, thereby leading to a distribution of travel distances that deviates from an idealized form associated with

uniform conditions. We may imagine similar effects on planar surfaces with nominally homogeneous roughness, but with local variations in roughness at the particle and slightly larger scale. However, whether these local effects could be distinguished from the inherent randomness of deposition is an open question. Note also that other processes may operate on hillslope surfaces such that stabilizing — or "smoothing" — influences do not need to be related just to rarefied particle motions as envisioned above. As an unusual example, "the impacts by small distal ejecta fragments... is the largest contributor to the

diffusive [topographic] degradation which controls the equilibrium [size-frequency distribution] of small craters" of the lunar maria (Minton et al., 2019). More generally, the formalism of generalized elastic models used to describe the macroscopic dynamics of fluctuating surfaces due to the competition between processes of surface roughening and relaxation (Pelletier and Turcotte, 1997; Turcotte, 2007) is now being extended to erosional landscapes (Schumer et al., 2017). Whether involving stabilizing effects or not, fluctuations in entrainment and deposition and accompanying variations in the land-surface state about

expected conditions are *just as important* as the expected conditions in characterizing surface behavior. Moreover, expected conditions may not represent a stable attractor in the sense of a mean-reverting process, akin to the stable basic state associated with slope-dependent soil creep (Furbish and Fagherazzi, 2001).

     Here is the key lesson. In the presence of noise driven processes with rarefied conditions, one must be cautious about predicting behavior in response to fixed continuum-like rates that do not acknowledge noise effects. Individual realizations

associated with these effects can involve rich behavior that is not anticipated from a simple deterministic perspective.





**Divergence with patchy entrainment:** Consider a similar situation in which entrainment events are Poisson in time and uniformly random in space, but the number of particles associated with each individual event is represented by an exponential distribution. Deposition and entrainment within an element $\Delta x$ are no longer a Poisson processes. Nonetheless, variations in the number of particles $n(t)$ in an element $\Delta x$ with time (not shown) qualitatively exhibit the same unstable behavior as Poisson

entrainment. Because individual events may involve numerous particles, the variability generally is larger. This reinforces the point made above, that a stable distribution of particle numbers with finite mean requires additional physics, for example, where the entrainment and deposition rates depend on the state $n(t)$.

## 4.3 Uncertainty with increasing scales

Here we consider uncertainty associated with rarefied particle motions on hillslopes viewed as a "slow" system, where changes

in hillslope configuration are largely imperceptible over the human time scale (Section 3.3). We highlight results from above, that with rarefied transport conditions our descriptions of the particle flux and its divergence pertain to ensemble conditions involving a distribution of possible outcomes, each realization being compatible with the controlling factors. When these factors change over time, individual outcomes reflect a legacy of earlier conditions that is influenced by the rate of change in the controlling factors relative to the intermittency of particle motions. The implication of this result together with preceding material

is that landform configurations reflect an inherent variability that is just as important as the expected (average) conditions in characterizing system behavior.

### 4.3.1 Ensemble expected conditions

We start by returning to a key starting point described in Section 4.2. Namely, despite the continuous forms of the entrainment rate $E_s(x,t)$, the probability density function $f_r(r;x,t)$ and the exceedance probability function $R_r(r;x,t)$ in the expressions

of the flux, Eq. (1), and the Exner equation, Eq. (2), these do not imply that the flux $q(x,t)$ and the rate of change in the land-surface elevation $\dot{\zeta}(x,t) = \partial\zeta(x,t)/\partial t$ may be considered as varying smoothly with space and time. Rather, for rarefied conditions these quantities are random variables. As a consequence the expressions Eq. (1) and Eq. (2) specifically represent ensemble expected conditions. Individual realizations may vary significantly from these expected conditions.

The simple Poisson processes described in the examples above suffice to illustrate the consequences of rarefied conditions,

where intermittency and patchiness add variability about expected conditions. Relative to the expected conditions, this variability may be large when viewed over small time scales. Only in the limit of a large number of particles with averaging over long time scales do predictions of the flux and its divergence approach expected (deterministic-like) values.

For any realization, the flux $q(x,t)$ and the rate $\dot{\zeta}(x,t)$ do not vary as continuously differentiable functions of position $x$ or time $t$. However, these quantities *are* well-defined continuously differentiable functions when applied to ensemble expected

conditions. In other words, when we write $\partial\zeta(x,t)/\partial t = -\partial q(x,t)/\partial x$ as a "model" of land-surface evolution in which $q(x,t)$ is specified by Eq. (1), we in fact are *imagining* a smoothly varying land-surface configuration that would occur if and only if $q(x,t)$ at all times coincides with the ensemble expected flux. A similar assessment applies to the entrainment form of the Exner equation, Eq. (2). Indeed, this description of the flux or the changing land-surface configuration is an idealization that





does not acknowledge noise effects. It is "continuum-like" in the sense that the described behavior proceeds in a continuously differentiable manner according to the (continuous) entrainment rate $E_s(x,t)$ convolved with either the smooth (continuous) exceedance probability $R_r(r;x,t)$ or the smooth probability density $f_r(r;x,t)$. In effect $E_s(x,t)$, $R_r(r;x,t)$ and $f_r(r;x,t)$ are treated as deterministic functions rather than probabilistically expected quantities.

More generally, Eq. (1) and Eq. (2) are probabilistic algorithms in which $E_s(x,t)$, $R_r(r;x,t)$ and $f_r(r;x,t)$ are statistically expected quantities. Each generated realization of $q(x,t)$ or $\dot{\zeta}(x,t)$ is entirely compatible with the controlling quantities, and is no less likely to occur than the expected value — although certain values are more probable than others according to the ensemble distribution of possible values. Thus, the ensemble distribution of possible values (Appendix D) is as important as the expected value in characterizing the behavior of the system. In the examples above involving Poisson events (delivery rate, entrainment rate), the behavior of $q(x,t)$, $n(x,t)$ or $\dot{n}(x,t)$ (or $\zeta(x,t)$ or $\dot{\zeta}(x,t)$) is not mean-reverting. The expected (average) state of the system therefore has no more mechanical significance than other state values.

    As outlined in Section 4.2.3, however, additional factors may provide stabilizing effects. Focusing on the rate $\dot{\zeta}(x,t)$, consider time varying conditions as the land surface changes (slope, surface roughness, etc.). One way to conceptualize this involves defining a zeroth-order configuration that changes slowly, and first-order fluctuations about the zeroth-order

state that change relatively rapidly. Following Sweeney et al. (2020), let $\zeta_0(x,t)$ denote a zeroth-order land-surface elevation and let $\zeta_1(x,t)$ denote a first-order deviation about the zeroth-order state. Then $\zeta(x,t) = \zeta_0(x,t) + \zeta_1(x,t)$ and $\dot{\zeta}(x,t) = \dot{\zeta}_0(x,t) + \dot{\zeta}_1(x,t)$. The zeroth-order rate $\dot{\zeta}_0(x,t)$ may be interpreted as representing ensemble expected conditions as described above. The first-order rate $\dot{\zeta}_1(x,t)$ then is akin to the behavior of an individual realization if this is conceptualized as a mean-reverting process. (The laboratory-scale experiments of Sweeney et al. (2020) indicate that this rate can be described as a

first-order autoregressive process, the discrete version of a mean-reverting Ornstein-Uhlenbeck process.) In effect, the slowly varying zeroth-order behavior is akin to the "climate" of the land surface and first-order fluctuations are akin to its "weather" (Sweeney et al., 2020).

    The analyses above focus on one-dimensional downslope transport. For completeness we note that the particle flux and its divergence more generally involve two-dimensional transport. For example, Williams and Furbish (2020) consider elements

of the two-dimensional forms of Eq. (1) and Eq. (2). They show how transverse diffusion of particles arises from particle-surface collisions during downslope travel, and how transverse motions influence the downslope particle flux. Clarifying the consequences of two-dimensional rarefied particle transport remains an interesting, open topic.

### 4.3.2   Legacy of realizations

The factors that control particle delivery rates and entrainment, as well as the conditions that influence particle motions and

deposition, change with time at different scales. For example, particle entrainment and surface-roughness texture associated with vegetal sediment capacitance may vary at fire recurrence time scales. Over longer time scales, continuing entrainment with downslope particle motions and deposition may contribute to changes in surface roughness and local land-surface slopes, thus changing the distribution of particle travel distances. At climate-change time scales, particle delivery rates to scree slopes may vary in relation to changing weathering rates and particle release from bedrock. Thus, we must acknowledge an adaptable



view of particle delivery and entrainment. Namely, an intermittent "event" during an episode of fire might be represented by the release of sediment from a vegetation capacitor. At the climate-change time scale, in contrast, an "event" may be viewed as consisting of the entirety of the release (or entrainment) of sediment associated with a fire and the period of post-fire recovery to vegetated conditions. Here we consider one element of the consequences of changes in the factors controlling particle motions,

in particular the possible mismatch between the time scale over which expected rates of delivery or entrainment change relative to the scale of intermittency in these rates.

Recall that the time-averaged flux eventually converges to the ensemble expected value with increasing elapsed time, and that the rate of convergence decreases with increasing intermittency in the delivery or entrainment of particles (Section 4.2.2). However, over intervals that are much shorter than the time required for convergence, the time-averaged flux in an individual

realization may differ significantly from the ensemble-averaged value. This is the same as saying that the number of particles moving past a position $x$ over a specified interval may be much different from the expected number, where the likely difference increases with increasing intermittency.

Consider for illustration the situation where particles are delivered intermittently to the top of a hillslope as a line source (Section 4.2.2). For illustration we specify the ensemble expected rate $\eta$ of events as an inhomogeneous rate that declines as

an exponential function with $e$-folding time $T_\eta$. A small value of $T_\eta$ implies a relatively rapid change in $\eta$ and a large value of $T_\eta$ implies a slow change. Recall that a decreasing value of $\eta$ coincides with increasing intermittency, and that the variability in particle numbers reaching a downslope position $x$ is more strongly influenced by this intermittency than by the expected number of particles per event. Then, a relatively large value of the dimensionless ratio $\eta T_\eta = T_\eta/\mu_w$ implies the expected delivery rate changes sufficiently slowly that the variability of individual realizations about ensemble expected conditions

is small. A relatively small value of $\eta T_\eta$ implies that the expected delivery rate changes faster than the rate at which the time-averaged flux converges to the expected rate (Figure 4). Thus, for decreasing $T_\eta$ and decreasing exceedance probability $R_x(x)$, individual realizations of the number of particles reaching position $x$ exhibit increasing variability about the ensemble expected values (Figure 6). Note that numerous other scenarios are possible. For example, a linear change in the expected rate (not shown) illustrates the same idea depicted in Figure 6. Specifically, these plots illustrate the growing effects of legacy in

previous controlling conditions with increasing intermittency and increasingly rarefied conditions. Namely, what occurs by chance under these conditions in the early part of the time series during rapidly changing expected conditions is inherited in later stages of the series as the rate of change in expected conditions decreases. It is only in the limit of vanishing intermittency relative to the $e$-folding time $T_\eta$ with large particle numbers near the source ($x = 0$, $R_x = 1$) that individual realizations track the ensemble expected conditions. With rapidly changing expected conditions, and far from the source, uncertainty in particle

numbers increases with time.

Specifically, and with reference to Figure 6d, if by chance during the early part of the series a relatively large number of events occurs, then this preconditions the total number $\hat{n}(x,t)$ as $\eta$ decreases with time $t$. The realization thus overshoots the ensemble expected state. If by chance during the early part of the series only a small number of events occurs, despite an initially large rate $\eta$, then this again preconditions the total number as $\eta$ decreases. The realization thus undershoots the

expected state.





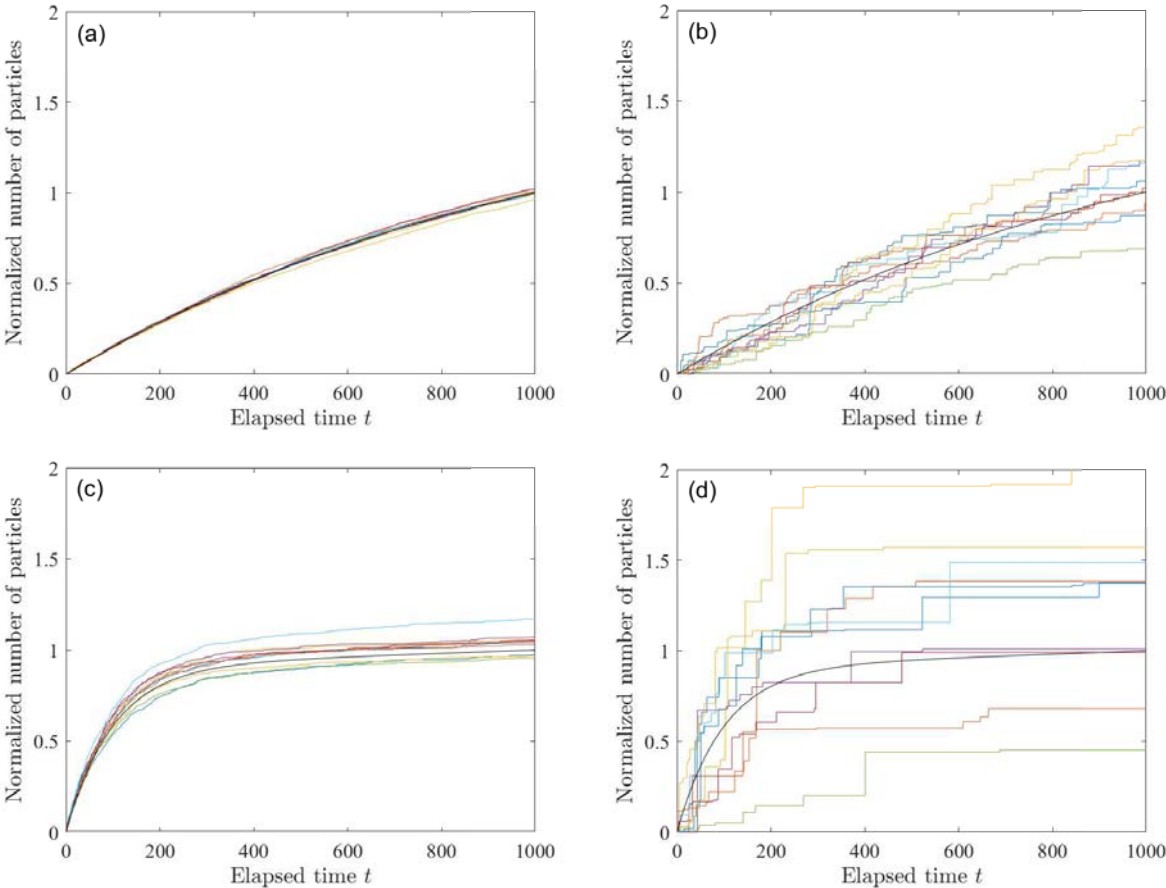

**Figure 6.** Plot of ten realizations (colored lines) of normalized number $\hat{n}(x,t)$ versus elapsed time $t$ showing increasing variability with $t$. Plots generated with $\eta(0) = 10$, $\eta(\infty) = 0.1$, $\mu_n = 1000$ and (**a**) $T_\eta = 1000$, $R_x = 1$, (**b**) $T_\eta = 1000$, $R_x = 0.01$, (**c**) $T_\eta = 100$, $R_x = 1$ and (**d**) $T_\eta = 100$, $R_x = 0.01$. Black line represents ensemble expected values of $\hat{n}(x,t)$.

Consider the slopes of the individual realizations in Figure 6 estimated by projecting lines of varying duration through different parts of the stepped curves. These slopes represent estimates of the particle flux. With increasingly rarefied conditions, notably when the expected rate $\eta$ rapidly changes, such estimated rates may be markedly different from the local expected rate associated with $\eta$. Further note that the idea of convergence of a time-averaged flux to an ensemble expected value with increasing averaging interval, as in the situation depicted in Figure 4, is not relevant in Figure 6. The ensemble expected value continuously changes over a time scale $T_\eta$ that is shorter than the time scale required for convergence, particularly with increasingly rarefied conditions.




## 5    Discussion and conclusions

In keeping with our philosophical objectives, we begin this section at a high level. This is to reinforce our view that it is important for growing efforts centered on probabilistic descriptions of sediment transport to include the philosophical underpinnings of this work within the conversation.

5       In 1943 Kurt Lewin first offered his oft quoted maxim that "there is nothing so practical as a good theory" (Lewin, 1943) for providing a framework to guide analyses of complex systems. This basic, lasting principle appears to resonate in many fields, particularly the social sciences (McCain, 2016). More recently, David Deutsch (2009, 2011) builds from ideas of Karl Popper to strongly argue for the essential guiding role of theory in the development of scientific explanation — that compelling explanations of natural phenomena are "theory laden." He forcefully rejects the idea of empiricism, that observation of the

world alone can suggest which ideas to adopt. In addition, Eugene Wigner provides an important elaboration of Lewin's maxim for the natural sciences. In his classic essay entitled, "The unreasonable effectiveness of mathematics in the natural sciences," Wigner (1960) notes that the triumph of physics resides in principles of invariance (Wigner, 1985) — that the laws of nature are invariant with any suitable transformation of space or time, thereby rendering them independent of initial conditions, position and history yet holding true for all time. He suggests that it is precisely the existence of this invariance that

gives us the confidence and inspiration — what he calls the "empirical law of epistemology" — for continuing the endeavor of discovery with growing complexity and uncertainty. Without this invariance, we would lose trust in our use of the laws of physics in different problems and settings — just as we would lose confidence and interest in playing the game of chess if the rules continually changed from one match to the next, precluding any gain in expertise from experience with fixed rules.

       Turning these ideas toward sediment systems, we suggest that the statistical mechanics framework outlined herein offers

a compelling strategy for examining particle motions and transport, particularly with rarefied conditions and in view of the uncertainty that goes with describing slow systems. This framework has two key elements that embody the points above. First, this framework is grounded in principles and methods for dealing with particle systems, continuum and rarefied, that have been rigorously scrutinized for more than a century. Its principles rest on Wigner's views of invariance, and its familiarity lends confidence for investing trust in its mechanical basis when examining unfamiliar problems outside of classical statistical

mechanics. Second, this framework embraces uncertainty at the outset in its use of probability. It offers established ways to formulate expressions of conservation, clear rules for counting and averaging particle states, and the foundational concept of a Gibbs ensemble. Again, we are inspired to invest trust in this formalism applied to unfamiliar systems. In particular, this framework points us in the right direction for examining the physics of rarefied particle motions on hillslopes, wherein we see the behavior of the particle system precisely for what it is — an unusual granular gas. The effort then consists of

elucidating a micro-view of the mechanical behavior of the particles during their downslope motions, which, when described probabilistically, leads to a macroscopic view of their collective (emergent) behavior.

       The theoretical analysis of particle motions involves threading together elements of statistical mechanics, concepts from granular gas theory, particle collision mechanics, and probability distribution theory (Furbish et al., 2020a, 2020c). Importantly, the analysis leans on the style of thinking of statistical mechanics while recognizing — as a delightfully challenging twist —





that it is not about simply adopting, off the shelf, theory and methods from this field. Instead, the work must be tailored to the transport process and scale of interest.

Each of the examples used in Section 4.1 to highlight the merits of a probabilistic description of particle motions and disentrainment — particle energy extraction, energy states and the Fokker-Planck equation, and the generalized Pareto distribution

as a maximum entropy distribution — represents a direct extension of established concepts in statistical mechanics as applied to both ordinary gases and granular gases. The analyses are not as straightforward as describing the behavior of ideal gas particle systems. Nonetheless, they nicely illustrate the transferability of basic principles, for example, the treatment of dissipative collisions as a random process, the value of appealing to a Gibbs ensemble as applied to a cohort of particles, and the use of an energetic cost to constrain the entropy maximization method. Thus, these examples illustrate elements of a coherent

statistical mechanics framework for describing sediment particle motions — that a mechanistic yet probabilistic analysis is possible. Moreover, the maximum entropy analysis specifically offers clarity on particle behavior that is not otherwise accessible. Namely, that all three forms of the generalized Pareto distribution are constrained in the same manner demonstrates that nothing special or unusual changes in the physics of disentrainment in the transition from the bounded form to the heavy-tailed form of the distribution in crossing isothermal conditions. The analyses thus rest on a solid foundation of statistical mechanics.

Nonetheless, it is essential that these results be challenged, and, if necessary, culled and replaced with fresh ideas.

With respect to consequences of rarefied versus continuum conditions, herein we focused on descriptions of the particle flux and its divergence (Section 4.2). Inasmuch as the particle delivery rate (as a line source) or the entrainment rate can be approximated as a Poisson or intermittent Poisson-like process, then the analysis clearly points to the idea that the flux or its divergence involves a distribution of possible outcomes, not just a single expected value — an idea that is decidedly different

from conventional continuum descriptions of these quantities. Note that the descriptions of the flux and its divergence do not depend on the results described above concerning the physics of particle motions. Indeed, the probabilistic nonlocal expressions of the flux and its divergence, Eq. (1) and Eq. (2), are independent of the form of the probability density $f_r(r; x, t)$ of particle travel distances $r$ and the associated exceedance probability function $R_r(r; x, t)$. Nonetheless, these expressions are firmly grounded in the methods of classical statistical mechanics, albeit specialized to sediment motions.

Here we reinforce the idea that Eq. (1) and Eq. (2) are probabilistic algorithms. For rarefied conditions the entrainment rate $E_s(x, t)$, although expressed as a continuously differentiable function of position $x$ and time $t$, is actually an expected rate constant. Particle entrainment "events" are decidedly discontinuous (Figures 3 and 5). Similarly, the continuous forms of the density $f_r(r; x, t)$ and the exceedance probability function $R_r(r; x, t)$ indicate that these represent ensemble expected conditions, not the outcome of any individual realization (Appendixes C and D). During an interval of time $\Delta t$ a finite number

$n$ of particles is entrained at the expected rate $E_s(x, t)$. This is equivalent to saying that a sample of size $n$ is drawn from the density $f_r(r; x, t)$. Such a sample, if plotted as a histogram of distances $r$, would have an irregular (discontinuous) form that only roughly mimics the smooth form of $f_r(r; x, t)$. Similar irregular histograms involving different values of $n$, no two alike, would occur during successive intervals $\Delta t$. As a consequence the flux $q(x, t)$ and the rate $\dot{\zeta}(x, t)$, although expressed as continuous functions, are decidedly discontinuous. All realizations are distinct, and none matches the ensemble expected state

(Figures 3, 4, 5 and 6).



As written, then, the flux $q(x,t)$ and the rate $\dot{\zeta}(x,t)$ are physically *imagined* quantities — as if the rate constant $E_s(x,t)$ represented a time-continuous "stream" of particle material distributed instantly and smoothly over space according to $f_r(r;x,t)$ or $R_r(r;x,t)$. It is only in this sense that Eq. (1) and Eq. (2) yield a single value of the flux or its divergence for specified controlling factors embodied in $E_s(x,t)$, $f_r(r;x,t)$ and $R_r(r;x,t)$. More precisely, these expressions describe how ensemble expected

values of the flux $q(x,t)$ and the rate $\dot{\zeta}(x,t)$ vary smoothly with position and time. That is, these ensemble expected quantities are well-defined continuously differentiable functions. Then, as noted in Section 4.3.1, if we write $\dot{\zeta}(x,t) = -\partial q(x,t)/\partial x$ as a "model" of land-surface evolution in which $q(x,t)$ is specified by Eq. (1), we in fact are describing an imaginary land-surface configuration that changes in a continuously differentiable manner if and only if $q(x,t)$ at all times coincides with the ensemble expected flux, neglecting any noise effects. Viewed in this manner, simulations of hillslope evolution based directly on Eq. (2)

(Furbish and Haff, 2010; Furbish and Roering, 2013; Doane et al., 2018) represent ensemble expected behavior. In contrast, simulations of hillslope evolution based on particle-based models (Tucker and Bradley, 2010; Bithell et al., 2014) represent individual realizations.

The examples involving Poisson or intermittent Poisson-like processes described in Sections 4.2 and 4.3 highlight the inherent variability that goes with noise driven processes, and point to an important consideration in interpreting landform config-

urations. Namely, for rarefied transport conditions a landform at any instant represents one of many possible realizations *for the same controlling factors*, whether fixed or varying with time. This view is distinctly different from the perspective offered by a smoothly varying deterministic model prediction that is based on fixed or slowing varying controlling factors without acknowledging noise effects that lead to a distribution of possible outcomes. The implication is that landform configurations reflect an inherent variability that is not simply attributable to perturbations about a deterministically expected state as in a

mean-reverting behavior (Furbish and Fagherazzi, 2001). This variability is just as important as an expected state in characterizing the landform behavior. Unfortunately, unlike fast systems, we cannot necessarily constrain the values of the controlling factors by direct measurements or in the manner of controlled experiments. We therefore must embrace the uncertainty that goes with rarefied transport conditions, and enjoy the weather of landforms as much as we do their imagined climate.

This perspective also induces us, while acknowledging consequences of the noisiness of rarefied systems, to examine the

dynamics of competition between roughening and smoothing processes (Schumer et al., 2017). As mentioned in Section 4.2.3, this may involve collective entrainment, the sediment capacitance of vegetation and other roughness elements, preferential entrainment and deposition in relation to surface geometry and roughness, effects of particle size sorting, or "smoothing" processes that are not related to rarefied particle motions per se. We suggest that there is value in taking cues from current work on noise driven bed load transport, including the coupling between moving particles and the streambed state (Ancey et al.,

2008; Ancey and Heyman, 2014; Pierce and Hassan, 2020). Whereas we have focused on local consequences of noisy delivery rates and entrainment, there is a need to systematically examine land-surface behavior in relation to rarefied particle transport.

In their examination of experimental time series of bed load flux, Ancey and Pascal (2020) provide an interesting lesson for considering slow systems. They show that for a noise driven process the time-averaged flux calculated from an individual realization (Figure 3 and 4) may differ significantly from the (known) sediment feed rate. We can imagine having information,

for example from a sediment deposit, that allows us to estimate the time-averaged sediment delivery rate associated with a



slow, noisy system. However, this estimated rate, representing an individual realization, may not coincide with the ensemble expected rate associated with the extant controlling conditions. In the absence of a high-fidelity time series of the delivery rate analyzed by re-sampling methods (Ancey and Pascal, 2020), the result is unavoidable uncertainty in this averaged rate.

We end with an anecdote to reinforce the starting point of this section. Last fall we wandered into Guilherme Gualda's graduate class on phase transformations in magmatic systems, and, to our delight, discovered on the chalkboard a derivation of the Boltzmann distribution of the energy states of atoms in a crystal lattice — complete with a pictorial rendering of the energy macrostates and microstates of a simple example system. The derivation continued with a description of the particle diffusion coefficient containing the Gibbs activation energy, as a direct consequence of the Boltzmann distribution, thence to the Arrhenius equation. We then enjoyed the discussion surrounding the idea that, in practice, one would experimentally determine

the diffusion coefficient for a real (i.e., not ideal) system rather than predict it from the statistical mechanics theory for specified atomic constituents and thermodynamic conditions. The students were quick! "So what is the value of the theory?" "Aha!," Gualda responded with delight, "the theory provides an unambiguous framework for interpreting our experimental data in view of the uncertainties of real systems! For example, in addition to providing a coherent, testable explanation of the phenomenon, the theory points to the appropriate functional form — the logical basis — of the expected relationship in curve fitting. This

in turn provides the basis of error assessment — either by classic propagation using the calculus or by Monte Carlo methods — which is particularly valuable given that experimental data often are sparse and of variable quality. And, it assigns clear meaning to estimated parameters for comparison with other work."

It is such a lovely, simple lesson: "There is nothing so practical as a good theory..." and, we would add in the case of sediment systems, "... that pays as much attention to fluctuations as it does to expected (mean) values."

*Data availability.* The data plotted in Figure 2 are available from sources described in Furbish et al. (2020b).

**Appendix A: Recent work on probabilistic elements of sediment motions and transport**

Here we offer a partial list of papers (81) representing recent work on probabilistic elements of sediment motions and transport in five topical areas. These papers contain numerous references to related work, including early probabilistic descriptions of transport and related material in the mathematics and physics literature. Under each heading we list the papers in the order of

their appearance.

Although these papers are just a sample, the relative numbers in the five areas accurately reflect the unevenness of efforts among these areas. The notable difference in efforts pertaining to rivers versus hillslopes is in part a direct reflection of the differences in our ability to observe and measure the transport processes. As noted in the main text, we know far more about bed load sediment transport in shear flows based on flume experiments than, say, soil particle transport and mixing associated

with bioturbation and granular creep.





**Bed load particle motions and transport:** Ancey et al. (2006), Ancey et al. (2008), Ancey (2010), Lajeunesse et al. (2010), Furbish et al. (2012a), Furbish et al. (2012b), Furbish et al. (2012c), Roseberry et al. (2012), Campagnol et al. (2013), Furbish and Schmeeckle (2013), Ancey and Heyman (2014), Heyman (2014), Heyman et al. (2014), Seizilles et al. (2014), Ancey et al. (2015), Fathel et al. (2015), Bohorquez and Ancey (2016), Fan et al. (2016), Fathel (2016), Fathel et al. (2016), Furbish et al. (2016a), Furbish et al. (2016b), Heyman et al. (2016), Furbish et al. (2017), Salevan et al. (2017), Ballio et al. (2018), Dhont and Ancey (2018), Lee and Jerolmack (2018), Ballio et al. (2019), Ancey (2020a), Ancey (2020b), Ancey and Pascal (2020), Ashlet et al. (2020), Chartrand and Furbish (2020), Pierce and Hassan (2020), Wu et al. (2020).

**Bed load tracer particle motions, including effects of particle-bed exchanges:** Hassan and Church (1991), Ferguson and Wathen (1998), Parker et al. (2000), Ferguson and Hoey (2002), Ferguson et al. (2002), Nikora et al. (2002), Wong et al. (2007), Schumer et al. (2009), Bradley et al. (2010), Ganti et al. (2010), Hill et al. (2010), Martin et al. (2012), Hassan et al. (2013), Phillips et al. (2013), Voepel et al. (2013), Martin et al. (2014), Pelosi et al. (2014), Phillips and Jerolmack (2014), Fathel et al. (2016), Bradley (2017), Hassan and Bradley (2017), Liu et al. (2019), Pierce and Hassan (2020b).

**Nonlocal sediment transport on hillslopes:** Foufoula-Georgiou et al. (2010), Furbish and Haff (2010), Gabet and Mendoza (2012), Furbish and Roering (2013), Doane (2018), Doane et al. (2018), Doane et al. (2019), Furbish et al. (2020a), Furbish et al. (2020b), Furbish et al. (2020c), Roth et al. (2020), Williams and Furbish (2020).

**Particle motions in soils, including tracer particles:** Furbish et al. (2009b), Furbish et al. (2018a), Furbish et al. (2018b), Furbish et al. (2018c), Gray et al. (2020).

**Rain splash transport:** Furbish et al. (2007), Furbish et al. (2009a), Dunne et al. (2010), Furbish et al. (2016b), Sochan et al. (2019).

## Appendix B: Divergence form of Exner equation

Consider the entrainment forms of the flux and the Exner equation, Eq. (1) and Eq. (2). Here we show that Eq. (2) is consistent with the divergence expressed as $c_b\dot{\zeta}(x,t) = -\partial q(x,t)/\partial x$ when the flux $q(x,t)$ is given by Eq. (1).

We start by writing $h(x,x') = E_s(x',t)R_r(x-x';x',t)$ so that Eq. (1) becomes

$$q(x,t) = \int_{-\infty}^{x} h(x,x')\,\mathrm{d}x'. \tag{B1}$$

Taking the derivative of Eq. (B1) and applying Leibniz's rule,

$$\frac{\mathrm{d}q(x,t)}{\mathrm{d}x} = \frac{\mathrm{d}}{\mathrm{d}x}\int_{-\infty}^{x} h(x,x')\,\mathrm{d}x'$$

$$= \int_{-\infty}^{x} \frac{\partial}{\partial x}h(x,x')\,\mathrm{d}x' + h(x,x)$$





$$= \int_{-\infty}^{x} \frac{\partial}{\partial x} [E_s(x',t)R_r(x-x';x',t)] \, \mathrm{d}x'$$

$$+ E_s(x,t)R_r(0;x,t). \tag{B2}$$

With $r = x - x'$ we observe that the operation $\partial/\partial x = \partial/\partial r$. Evaluating the derivative in Eq. (B2), noting that $f_r(r;x,t) = -\mathrm{d}R_r(r;x,t)/\mathrm{d}r$ and $R_r(0;x,t) = 1$, then yields

$$\frac{\mathrm{d}q(x,t)}{\mathrm{d}x} = - \int_{-\infty}^{x} E_s(x',t)f_r(x-x';x',t) \, \mathrm{d}x'$$

$$+ E_s(x,t). \tag{B3}$$

Reverting to partial notation the entrainment form of the Exner equation, Eq. (2), then follows from $c_b \dot{\zeta}(x,t) = -\partial q(x,t)/\partial x$.

Consider an alternative formulation. With $r = x - x'$ we note that $\mathrm{d}x' = -\mathrm{d}r$. We now use this change of variable to rewrite Eq. (1) as

$$q(x,t) = \int_{0}^{\infty} E_s(x-r,t)R_r(r;x-r,t) \, \mathrm{d}r. \tag{B4}$$

This form of the convolution may be expanded as a Taylor series to show that the flux consists of advective and diffusive 15 parts so long as the integral of $R_r(r;x,t)$ converges (Furbish and Roering, 2013; Furbish et al., 2017). In turn we use the communicative property of convolutions to write Eq. (B4) as

$$q(x,t) = \int_{-\infty}^{x} E_s(r,t)R_r(x-r;r,t) \, \mathrm{d}r, \tag{B5}$$

which has the same for as Eq. (1). We then take the derivative of Eq. (B5) and proceed as above to obtain the entrainment form of the Exner equation, Eq. (2).

**Appendix C: Rarefied versus continuum conditions**

The material in this appendix is mostly extracted directly from Appendix A in Furbish et al. (2018c). Our aim is to further illustrate the significance of rarefied versus continuum conditions, and the interpretation of the Fokker-Planck equation applied to these conditions. We focus on the familiar example of Brownian motion, the initial formal description of which is separately attributable to Einstein (1905) and von Smoluchowski (1906). For additional background, Schumer et al. (2009) provide a 25 particularly clear description of the Lagrangian perspective of particle motions and its relation to the Eulerian perspective of particle behavior as embodied in the Fokker-Planck equation.



Earth **Surface**
**Dynamics**
Discussions



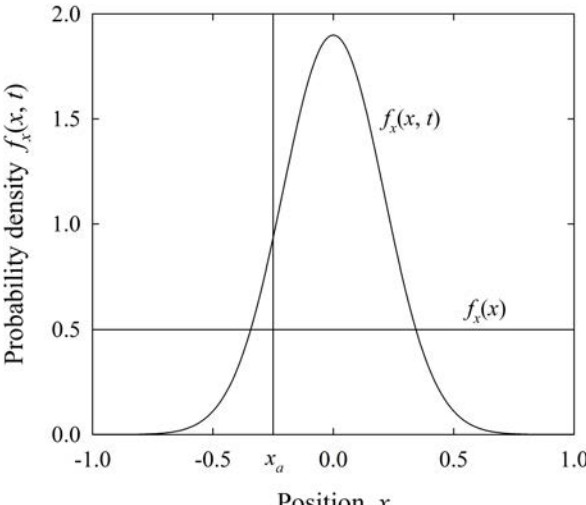

**Figure C1.** Plot of coordinate position $x$ of particle undergoing a random-walk motion showing Gaussian distribution $f_x(x,t)$ of expected positions at time $t$ as the solution, Eq. (C2), of the Fokker–Planck equation, and the actual (example) particle position $x = x_a$, and uniform steady-state distribution $f_x(x) = 1/2$ for a bounded domain such that $-1 < x < 1$. Figure reproduced from Furbish et al. (2018c).

With reference to Figure C1, let $x$ denote a coordinate along which Brownian particles take one-dimensional random walks, where $x$ extends indefinitely in the positive and negative directions about the origin $x = 0$. Suppose that a particle starts at the origin at time $t = 0$, and with equal probability moves in the positive or negative direction during successive small intervals $dt$. By the definition of a random walk, the motion of the particle — specifically its expected position $x$ after an interval of

time $t > 0$ — can be predicted only in a probabilistic sense. Namely, letting $f_x(x,t)$ denote the probability density function of possible positions $x$, then this density satisfies a Fokker-Planck equation involving only its diffusion term:

$$\frac{\partial f_x(x,t)}{\partial t} = k_2 \frac{\partial^2 f_x(x,t)}{\partial x^2},$$
(C1)

where the particle diffusivity $k_2$ is assumed to be constant. The solution of Eq. (C1) is the Gaussian distribution with mean $\mu_x = 0$, namely,

$$f_x(x,t) = \frac{1}{\sqrt{4\pi k_2 t}} e^{-x^2/4k_2 t}.$$
(C2)

For this highly rarefied system involving a single particle, we can only offer probabilistic predictions of its position at time $t$. For example, we may confidently state that with probability $p = 1/2$ the particle is either at a position $x < 0$ or at a position $x > 0$. Or we may state that with probability $p \approx 0.68$ the particle is within the domain defined by plus one and minus one standard deviations about the mean position, namely, $-\sqrt{2k_2 t} < x < +\sqrt{2k_2 t}$. For this single-particle system (realization),

the actual particle position $x_a$ is represented by a Dirac distribution $\delta(x_a - x, t)$ (Figure B1), but this cannot be predicted deterministically (Schumer et al., 2009).





Earth **Surface**
**Dynamics**
Discussions

Let us now imagine an arbitrarily great number $N$ of identical, independent particles that start at the origin $x = 0$ at time $t = 0$, each undergoing a random walk during $t > 0$. When viewed together, the distribution of these particles at time $t = 0$ is given by the Dirac distribution, namely, $f_x(x, 0) = \delta(x)$. At any time $t > 0$ these particles are distributed according to Eq. (C2). That is, because $N$ is arbitrarily large, the proportion of particles within any small interval $x$ to $x + \mathrm{d}x$ closely matches what is

5    predicted by Eq. (C2), namely $f_x(x, t)\mathrm{d}x$, such that in the limit of $\mathrm{d}x \to 0$ the actual distribution of positions $x$ varies smoothly (continuously) and converges to Eq. (C2) (Figure C2). In contrast to the highly rarefied single-particle system in the previous

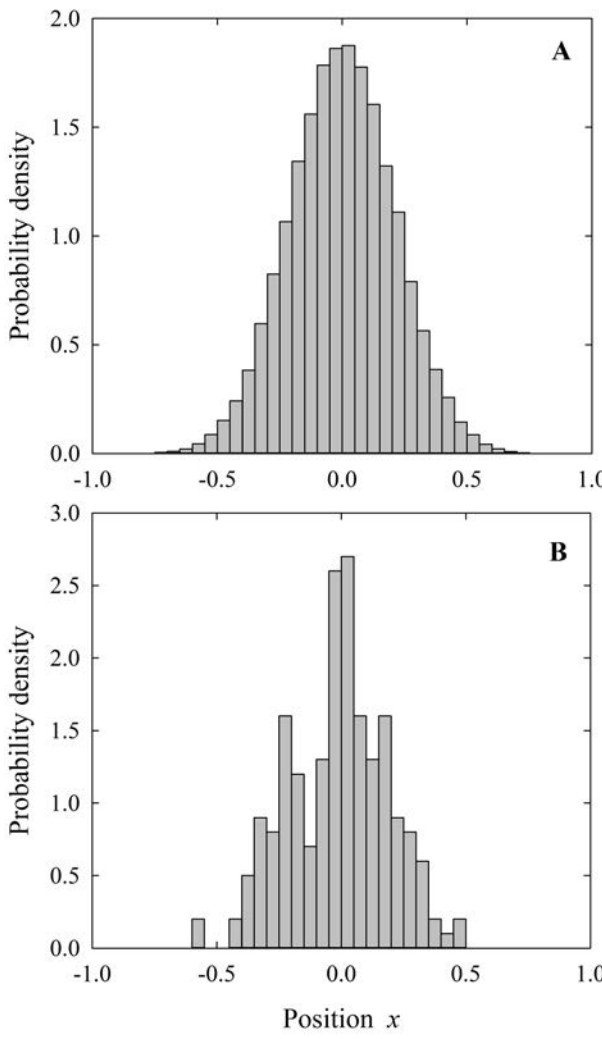

**Figure C2.** Histograms of particle positions $x$ at time $t$ for one system showing that (**a**) with a great number $N$ of particles representing a continuum condition this histogram converges to the smooth Gaussian distribution in Figure B1 as $\mathrm{d}x \to 0$; in this example $N = 100000$, and (**b**) with a modest number of particles representing a rarefied condition this histogram is irregular and discontinuous; in this example $N = 200$. Figure reproduced from Furbish et al. (2018c).





example, we may thus assume that this great number of particles, occurring in one system (realization), satisfies the continuum hypothesis. Nonetheless, upon randomly selecting a single particle from this system, we still can only offer probabilistic predictions of its position at time $t$ — as in the example above involving a system with a single particle. Moreover, note that the continuous distribution of positions $x$ realized at time $t$ for this one system involving a great number $N$ of particles is

identical to the distribution that would be realized upon pooling the $x$ positions at time $t$ associated with a great number $N$ of independent systems, each involving a single particle.

Now select a system with a modest number $N$ of particles such that conditions are rarefied. By this we mean that, after some time $t$, the actual distribution of particle positions $x$ is at best represented by an irregular histogram that roughly appears Gaussian but is decidedly discontinuous (Figure C2). Moreover, any realization involving $N$ particles possesses a similar

irregular form at time $t$, and no two are the same. In effect, each realization represents a sample of size $N$ drawn from an imagined population represented by Eq. (C2). Also note that each realization involving $N$ particles at time $t$ is the same as $N$ realizations, each involving a single particle, when viewed collectively at time $t$.

Let us now consider a great number $N_e$ of independent but nominally identical systems — an ensemble — at any fixed time $t$, where each system contains $N$ particles, large or small. We now wish to describe the ensemble-expected conditions.

To envision this, consider any small interval $x$ to $x + \mathrm{d}x$. If $N = 1$ as in the first example above, then $f_x(x,t)\mathrm{d}x$ is just the proportion of the $N_e$ systems containing a particle within $x$ to $x + \mathrm{d}x$ at time $t$. Note that this is identical to the result above involving an individual system containing a great number $N = N_e$ of particles. If instead each system involves a great number $N$ of particles, then $f_x(x,t)\mathrm{d}x$ simply becomes the expected proportion of the $N$ particles within $x$ to $x + \mathrm{d}x$ at time $t$, where the expectation is calculated over the $N_e$ systems. And note that this outcome is identical to the proportion of $N \times N_e$

independent systems, each involving a single particle, which contain a particle within $x$ to $x + \mathrm{d}x$ at time $t$. In either case, the expected proportion within the interval is the same. Moreover, we reach the same conclusion in considering a great number $N_e$ of systems, each involving a modest number $N$ of particles. Thus, when calculated over a great number of systems for all intervals $\mathrm{d}x$, then in the limit of $\mathrm{d}x \to 0$, the continuous function, Eq. (C2), is retrieved. The key points are these: first, whether $N$ is relatively small (representing a rarefied condition) or $N$ is large (representing a continuum condition), the ensemble-

expected behavior represented by Eq. (C2) applies equally to both conditions in a probabilistic sense. Second, if $N$ is small, then Eq. (C2) represents the ensemble-expected behavior, not the actual behavior of any one system (realization); if $N$ is large, then the actual behavior of the system is expected to converge to the smooth ensemble behavior represented by Eq. (C2).

To complete the picture, suppose that the $x$ domain in Figure C1 is bounded such that $-1 < x < 1$. Particles that reach these boundaries are "reflected" and remain within the domain, continuing their random walks. In the limit of $t \to \infty$, the

probability density of particle positions $x$ reaches a steady-state form, that is, $\partial f_x(x,t)/\partial t \to 0$ such that $f_x(x,t) \to f_x(x)$. In this limit, Eq. (C1) becomes $\mathrm{d}^2 f_x(x)/\mathrm{d}x^2 = 0$. Moreover, the probability flux $q_x = -k_2 \mathrm{d}f_x(x)/\mathrm{d}x = 0$ at all positions $x$, which means that $\mathrm{d}f_x(x)/\mathrm{d}x = 0$. These constraints together with the fact that the distribution $f_x(x)$ must integrate to unity yield the result that $f_x(x) = 1/2$ over the bounded domain (Figure C1). That is, the expected distribution $f_x(x)$ is uniform. As with the unsteady problem described above, a modest number $N$ of particles representing rarefied conditions in any one

realization is at best represented by an irregular histogram that roughly appears uniform but is decidedly discontinuous (Figure



C3). Moreover, at an arbitrary later time, the resulting distribution (histogram) would be just as irregular; it does not become

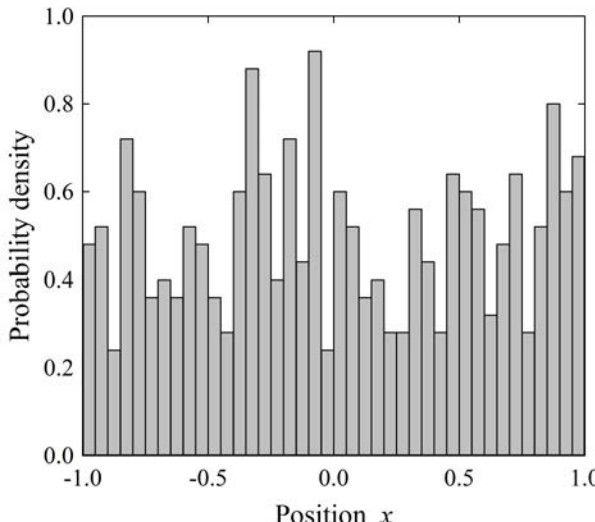

**Figure C3.** Histogram of particle positions $x$ at time $t \to \infty$ for one system showing that with a modest number of particles representing a rarefied condition this histogram is irregular and discontinuous; in this example $N = 500$. Figure reproduced from Furbish et al. (2018c).

smoother with increasing time. As above, the expected continuous steady-state distribution is retrieved when expected values are calculated over a great number $N_e$ of systems.

With respect to developments in the text, the Fokker–Planck equation describes the time evolution of the probability density
$f_x(x,t)$. The formulation does not assume either rarefied or continuum conditions. It is indifferent to these conditions, yet equally applicable to both.

## Appendix D: Ensemble expected conditions of a Poisson process

Here we provide an explicit definition of an ensemble expected value or state and its relation to a time-averaged value. Recall that the Poisson distribution, Eq. (25), describes the probability that $n$ events will occur within a specified interval of time $t$
when these events occur randomly with a fixed rate constant $\eta$ [T$^{-1}$]. Here we write this distribution as

$$f_n(n,t) = \frac{(\eta t)^n}{n!} e^{-\eta t},$$  (D1)

with mean $\mu_n = \eta t$ and variance $\sigma_n^2 = \eta t$. As written, Eq. (D1) looks like a description of the time evolution of a system involving Poisson events. However, note that for an individual realization of a Poisson process over a specified interval $(0,t)$ there is only one outcome, that is, precisely $n$ events. As usually presented, we are to imagine such an interval of time and use
Eq. (D1) to assess the likelihood that $n$ events will occur during the interval. Here we alternatively consider an ensemble of systems.



Let us imagine, as did Gibbs (1902), a great number $N_e$ of nominally identical but independent systems (an ensemble), each subject to the same physics of random events characterized by the rate constant $\eta$. (One may consider each realization plotted in Figure 3, for example, as representing the outcome of one system in the ensemble.) Let the subscript $i = 1, 2, 3, \ldots, N_e$ denote the individual systems composing the ensemble. We now imagine starting each system at time $t = 0$ with the initial conditions,

$$f_n(0,0)_i = 1 \qquad \text{and} \qquad f_n(n,0)_i = 0, \quad n \geq 1. \tag{D2}$$

That is, at time $t = 0$ each of the $N_e$ systems has a value of unity at $n = 0$ and a value of zero for all $n \geq 1$. Taking the ensemble expectations,

$$f_n(0,0) = \frac{1}{N_e} \sum_{i=1}^{N_e} f_n(0,0)_i = \frac{N_e}{N_e} = 1 \qquad \text{and} \tag{D3}$$

$$f_n(n,0) = \frac{1}{N_e} \sum_{i=1}^{N_e} f_n(0,0)_i = \frac{0}{N_e} = 0, \quad n \geq 1. \tag{D4}$$

Thus, $f_n(0,0)$ is just the proportion of the $N_e$ systems with $n = 0$ events and $f_n(n,0)$ is the proportion of the $N_e$ systems with $n \geq 1$ events at time $t = 0$.

Now, at any time $t > 0$ each system has precisely $n$ events with probability one. Those that have $n = 0$ events are represented by

$$f_n(0,t)_i = 1 \qquad \text{with} \qquad f_n(n,t)_i = 0, \quad n \neq 0, \tag{D5}$$

those that have $n = 1$ events are represented by

$$f_n(1,t)_i = 1 \qquad \text{with} \qquad f_n(n,t)_i = 0, \quad n \neq 1, \tag{D6}$$

those that have $n = 2$ events are represented by

$$f_n(2,t)_i = 1 \qquad \text{with} \qquad f_n(n,t)_i = 0, \quad n \neq 2, \tag{D7}$$

and so on for systems with $n = 3, 4, 5, \ldots$ events. Now let $N(n)$ denote the number of systems with $n$ events. Taking ensemble expectations,

$$f_n(0,t) = \frac{1}{N_e} \sum_{i=1}^{N_e} f_n(0,t)_i = \frac{N(0)}{N_e}, \tag{D8}$$

$$f_n(1,t) = \frac{1}{N_e} \sum_{i=1}^{N_e} f_n(1,t)_i = \frac{N(1)}{N_e} \tag{D9}$$





and so on for systems with $n = 2, 3, 4, \ldots$ events. That is, each probability $f_n(n, t)$ for $n = 0, 1, 2, \ldots$ is just the proportion of the $N_e$ systems with $n$ events at time $t$. More generally,

$$f_n(n, t) = \frac{1}{N_e} \sum_{i=1}^{N_e} f_n(n, t)_i = \frac{(\eta t)^n}{n!} e^{-\eta t}, \quad n \geq 0, \tag{D10}$$

which is our starting point. At this juncture the form of Eq. (D10) now may be interpreted as describing the time evolution of the ensemble distribution $f_n(n, t)$ (Figure D1). To reiterate, upon starting each member of the ensemble at $t = 0$ according to

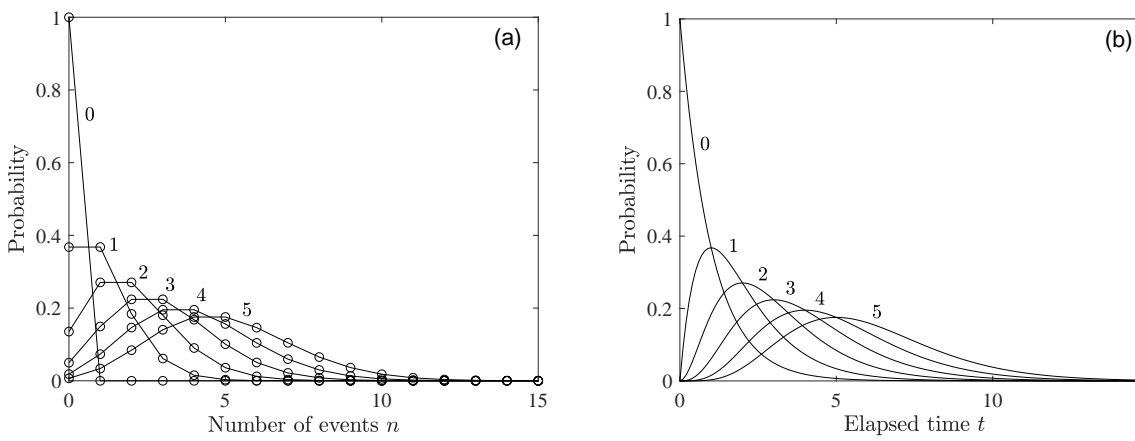

**Figure D1.** Plot of time evolution of (**a**) Poisson ensemble distribution $f_n(n, t)$ of the proportion $N(n)/N_e$ of systems with $n$ events at time $t = 0, 1, 2, 3, 4$ and 5, and (**b**) proportion $N(n)/N_e$ of systems with $n = 0, 1, 2, 3, 4$ and 5 events. Plots are based on $\eta = 1$. Lines connecting the probability values (circles) in (**a**) are to aid visualization of successive states of the discrete distribution and do not imply the presence of non-integer values of $n$.

Eq. (D2) then letting time proceed, Eq. (D10) describes the distribution of the values of $n$ associated with the $N_e$ members of the ensemble when viewed at any instant. Whereas at time $t = 0$ all members of the ensemble have $n = 0$ events with a probability of one, the proportion $f_n(0, t)$ decays as $e^{-\eta t}$ with $t > 0$. The proportion $f_n(1, t)$ initially grows, reaches a peak, then decays as $\eta t e^{-\eta t}$. The proportion $f_n(2, t)$ likewise initially grows, reaches a peak, then decays as $(1/2)(\eta t)^2 e^{-\eta t}$. This

10 pattern continues for proportions involving $n = 3, 4, 5, \ldots$.

The essential idea is this. For any individual system there is one possible outcome $n$ at time $t$ with probability given by Eq. (D1). For an ensemble of systems all possible outcomes $n = 0, 1, 2, 3, \ldots$ exist at time $t$ in the proportions given by Eq. (D1). In fact, these ensemble proportions constitute the formal, classic definition (Hájek, 2012) of the probabilities $f_n(n, t)$ of $n$ events at time $t$.

15 Consider the situation where particles are delivered as a line source ($x = 0$) at the expected rate $\eta$. In this problem the ensemble expected particle flux at position $x > 0$ is $R_x(x)\eta$, which by definition is the expected time-averaged flux for any averaging interval $t$. This generally is not the same as the time-averaged flux of any realization. Namely, if $n_i(x, t)$ denotes the total number of particles moving past $x$ during the interval $t$ for the $i$th realization, then the time-averaged flux is $n_i(x, t)/t$.





This time average converges to the expected rate $R_x(x)\eta$ only in the limit of $t \to \infty$. To gain a sense of this convergence we may consider the value of the time-averaged flux coinciding with one standard deviation in the expected number of particles moving past $x$ during $t$, which is $\sqrt{R_x(x)\eta t}/t = [R_x(x)\eta/t]^{1/2}$. This suggests that the flux converges as $\sim t^{-1/2}$, which is slower than exponential convergence.

Whereas the ensemble distribution represented by Eq. (D10) evolves with time $t$, for completeness we comment here on the idea of a "stable," or stationary, ensemble distribution that is independent of time. The geometric distribution associated with the M/M/1 queuing problem with $\eta_I < \eta_E$ (Section 4.2.3) is a stable distribution. Individual realizations of $n(t)$ may fluctuate over the domain $n = 0, 1, 2, 3, ...$, but at any instant the ensemble distribution $f_n(n)$ is independent of time. Similarly, the binomial and negative binomial distributions $f_{n_a}(n_a)$ describing the number $n_a$ of active particles (Ancey et al. 2008; Heyman, 2014;

Heyman et al., 2014) are stable distributions. The uniform distribution associated with a Brownian particle as described in Appendix D is a stable (time-independent) distribution. The exponential distribution of particle velocities described by Furbish et al. (2012b), Furbish and Schmeeckle (2013) and Fathel et al. (2015) is considered to be a time-independent ensemble distribution.

*Author contributions.* The reported work represents an intellectual co-conspiracy between the authors growing from the PhD work of THD.

*Competing interests.* We have no competing interests.

*Acknowledgements.* We acknowledge support by the U.S. National Science Foundation (EAR-1420831 and EAR-1735992). We appreciate continuing discussions with Peter Haff and Sarah Williams concerning descriptions of Earth-surface systems. Shawn Chartrand and Tom Dunne offered useful reactions to an earlier draft.





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
