# Peer review of "Rarefied particle motions on hillslopes: 4. Philosophy"

_Earth Surface Dynamics, 2020_

## Short Comment (SC1) · 15 Jan 2021

The present suite of papers focuses on the special case of rarefied grain collisions, where particle-surface interactions dominate mutual interactions between in-flight grains. Together these contributions illustrate the progression from micro-dynamics to macro-scale observation using laboratory and field-based measurements. This fourth paper on "philosophy" provides an extended discussion of the framework and strategy of the technical analysis, which is based on an analogy with statistical mechanics. I suspect that many geomorphologists who deal with sediment transport and landscape evolution, if they scan through recent ESD postings, will pause and poke only briefly, if at all, at this set of papers with its heavy reliance on concepts such as maximum entropy distribution and Fokker-Planck equations. In my view that would be a mistake.

These papers open up many vistas on grain transport that might otherwise be missed. I would suggest the reader focus first on this final paper in the series. It alerts readers to a range of interesting problems that are difficult either to state or to resolve in the language of familiar continuum methods of analysis, but which can be usefully approached from a foundation of statistical mechanics. These might include problems in risk assessment for example, in which specific outcomes, even when many particles are involved, need not be reliably close to "average behavior" (the distinction made by the authors between granular "weather" and granular "climate"). Generic grain transport problems that might benefit by paying greater attention to statistics of the underlying particle dynamics include effects of granular size, shape, density, friction coefficients and elastic moduli on erosion, sedimentation and sorting of particles. To aid accessibility and to draw readers in, it might be useful to add in abbreviated form some of the philosophical or "framing" content of the last paper into the early part of the text of the first paper of the series. This could help clarify at the outset the overall unity and import of the overall body of work, which I hope ESD will promote to full publication.

---

## Referee Comment (RC1) · Joris Heyman (Referee) · 1 Feb 2021

4) Philosophy :

The fourth paper present a general discussion on probabilistic approach to rarefied particle motions. It correctly points the generality of such approach, and shows how continuum equation of motion extend (within some subtle extra terms) to ensemble average quantities or probability distributions, even when the instantaneous particle flux is strongly intermittent.

While I completely agree with this viewpoint, and I believe the paper has its importance for the community, I am not sure how this relates specifically to the hillslope motions. Indeed, the use of ensemble averaging/probabilistic description to describe

rarefied gas, bedload, or avalanches, and the scale dependence of fluctuations, is a much more general discussion that could fit in a standalone study, with dedicated title. Indeed, the 4th papers format dilutes in my sense the distinct messages the authors convey. Nevertheless, if the editors and reviewers think the inclusion of this paper as a companion paper is justified I will not argue against this.

One minor comment is the following. The authors point 2 equivalent probabilistic viewpoints, the Fokker-Planck equation (the linearization of the master equation) and the maximum entropy approach, originating from statistical physics (they discussed in the 1st and 3rd companion paper). In the discussion, I would include a third way, the Poisson representation [1], which has the attracting characteristic of being exactly equivalent to the Master Equation, while leading to continuous, analytically tractable PDEs. This approach, developed by Gardiner, can be used [1,2] to compute the exact particle number pdf and correlations from basic entrainment/disentraiment rules, without requiring a "small" noise or Kramer-Moyal expansion that assume a large number of particles. As pointed by Gardiner, it has the potential to describe "low density-high fluctuations" states of granular gases, for which large deviations play an important role. A mention of such alternative could be relevant.

1 Gardiner, C. W. (1985). Handbook of stochastic methods (Vol. 3, pp. 2-20). Berlin: springer. 2 Ancey, C., & Heyman, J. (2014). A microstructural approach to bed load transport: mean behaviour and fluctuations of particle transport rates. Journal of Fluid Mechanics, 744, 129-168. 3 Heyman, J., Ma, H. B., Mettra, F., & Ancey, C. (2014). Spatial correlations in bed load transport: Evidence, importance, and modeling. Journal of Geophysical Research: Earth Surface, 119(8), 1751-1767.

Please also note the supplement to this comment:
https://esurf.copernicus.org/preprints/esurf-2020-101/esurf-2020-101-RC1-supplement.pdf

**ESurfD**

---

## Referee Comment (RC2) · Anonymous Referee #2 · 11 Mar 2021

In the 4th paper of the companion papers, the authors explain their scientific strategy and philosophical viewpoint for attacking the highly important and yet unsolved problem of soil transport on rough landscapes over a wide range of space and time scales. The paper covers a very broad range of significant issues, that all of them are pertinent to soil and sediment transport in various settings (and not necessarily on hillslopes). The work also sets a high bar and a valuable example for future research in this and adjacent fields. It further provides variety of ideas and problems that can become future research topics by other researchers and inspire much needed work to explore and illuminate the physics and mechanics of soil and sediment transport. At this point, I would like to add that the paper is also heavy in arguments based on statistical mechanics and kinetic theory of gases. This reviewer is not an expert in either of these fields, and they only have an introductory knowledge to follow the argument. Therefore, I encourage the editors or interested readers to further evaluate the statistical mechanics-based arguments presented in the paper by themselves, or by seeking additional input from experts in those fields (I noticed that editors might have already sought feedback from experts in those areas). I support the publication of this part of the companion papers, after the authors have revised the paper to address my mostly minor comments and questions below:

- This might have been discussed in the earlier parts of the companion papers — which I didn't have the resources to study in detail, before completing my review task —, however, I would appreciate it if the authors can elaborate in this part, the conditions under which particle transport can be considered rarefied. I think providing a quantitative statement, potentially a dimensionless number or a metric that can be measured in experiment/simulation/field, would be very valuable.

- Related to my previous comment, do the authors think or advocate that all soil transport (or all that matters for soil transport phenomenon) is in rarefied condition? I have an opposing view, but at this point, it maybe just a matter of my misunderstanding. I try to explain my view in the next few sentences. I think of soil transport as a continuum — i.e., coexisting and gradually transitioning between them — of modes of transport and transport conditions. I agree that rarefied condition is a big part of that. However, as the ratio of particle per volume (packing fraction or solid fraction) increases as we get closer (from air) to (hillslope) surface, the transport condition becomes closer to the dense flow or dense particle transport. My view is in part informed by the experiments by Houssais et al (2015), where the authors show that in the case of fluvial sediment transport, there are three regimes (suspended particles, bedload, and creep; see their Fig. 1D). Some may argue that these are just two regimes by considering suspended transport as a part of the bedload, from the viewpoint of sediment transport, or by considering creep as part of the bedload, from the viewpoint of granular flow and physics; but in any case, there are at least two regimes there. How can the framework described here be applied to, or otherwise remain relevant to, such conditions in the lab and field, where the entire system may not be in the rarefied condition? Would you advocate for ignoring the contributions of the dense regimes, or otherwise suggest to readers to focus for future research on the rarefied condition of soil transport? Do you consider the contributions of the dense flow part of the transport as *solved* with the existing heuristic equations and relations, especially if they cannot be explained or adequately modeled in the rarefied framework described in the companion papers?

- The behavior of materials (and not general phenomenology of transport) in the dense regime is highly sensitive to size and size distribution, shape, etc of granular materials. I think there is an over emphasis in the manuscript on the probabilistic approach to the problem (which I feel like to be very useful for rarefied transport condition). However, to the degree that the contributions of dense flow regime remain relevant to soil transport, I would favor a slightly more balanced research and scientific strategy, where there is enough space to explore and investigate the physics and mechanics of dense regime. I would also argue for integrating more of reductionist studies, e.g., on the physics of mixing using the laboratory or simulation data on the feasible spatiotemporal scales, and less worry in the first instance about the uncertainty of those measurements for application to the geological space and time. The authors have mentioned this at some points in the paper (the part about discrete element modeling simulations or other first principle or ab initio studies that can be accompanied by carefully crafted experiments and theory development), but I think those avenues might be worthy of some more attention, especially in this philosophical part of the companion papers.

- This a very minor comment. On page 19, ~line #30, the authors put close to each other, what is called "nonlocal behavior/rheology" in the dense granular flow research community, and the nonlocal transport in the sediment transport research community. First, this gives me the impression that the authors consider dense granular/particle flow to be in the rarefied condition. I don't think this is a correct view, but I am happy to learn more about the authors viewpoint on that, so it would be helpful if they can clarify on this matter. Second, I think there is still ongoing debate related to the dense granular flow behavior, and whether it should be called "nonlocal behavior" or "nonlocal rheology". This issue is not yet settled in the granular and complex fluids research communities, and the resembles between the two terms (nonlocal rheology and nonlocal transport) may cause confusion or concern for some readers.

**References**

Houssais, M., Ortiz, C. P., Durian, D. J., & Jerolmack, D. J. (2015). Onset of sediment transport is a continuous transition driven by fluid shear and granular creep. Nature communications, 6(1), 1-8.

---

## Author Comment (AC1) · 25 Mar 2021

The comment was uploaded in the form of a supplement:
https://esurf.copernicus.org/preprints/esurf-2020-101/esurf-2020-101-AC1-supplement.pdf